# Oomycete small RNAs bind to the plant RNA-induced silencing complex for virulence

Florian Dunker[1], Adriana Trutzenberg[1], Jan S Rothenpieler[1], Sarah Kuhn[1], Reinhard Pröls[2], Tom Schreiber[3], Alain Tissier[3], Ariane Kemen[4], Eric Kemen[4], Ralph Hückelhoven[2], Arne Weiberg[1]*

[1]Faculty of Biology, Genetics, Biocenter Martinsried, LMU Munich, Martinsried, Germany; [2]Phytopathology, School of Life Sciences Weihenstephan, Technical University of Munich, Freising, Germany; [3]Department of Cell and Metabolic Biology, Leibniz Institute of Plant Biochemistry, Halle, Germany; [4]Center for Plant Molecular Biology, Interfaculty Institute of Microbiology and Infection Medicine Tübingen, University of Tübingen, Tübingen, Germany

**Abstract** The exchange of small RNAs (sRNAs) between hosts and pathogens can lead to gene silencing in the recipient organism, a mechanism termed cross-kingdom RNAi (ck-RNAi). While fungal sRNAs promoting virulence are established, the significance of ck-RNAi in distinct plant pathogens is not clear. Here, we describe that sRNAs of the pathogen *Hyaloperonospora arabidopsidis*, which represents the kingdom of oomycetes and is phylogenetically distant from fungi, employ the host plant's Argonaute (AGO)/RNA-induced silencing complex for virulence. To demonstrate *H. arabidopsidis* sRNA (*Hpa*sRNA) functionality in ck-RNAi, we designed a novel CRISPR endoribonuclease Csy4/GUS reporter that enabled in situ visualization of *Hpa*sRNA-induced target suppression in Arabidopsis. The significant role of *Hpa*sRNAs together with *At*AGO1 in virulence was revealed in plant *atago1* mutants and by transgenic Arabidopsis expressing a short-tandem-target-mimic to block *Hpa*sRNAs, that both exhibited enhanced resistance. *Hpa*sRNA-targeted plant genes contributed to host immunity, as Arabidopsis gene knockout mutants displayed quantitatively enhanced susceptibility.

*For correspondence:
a.weiberg@lmu.de

Competing interests: The authors declare that no competing interests exist.

## Introduction

Plant small RNAs (sRNAs) regulate gene expression via the Argonaute (AGO)/RNA-induced silencing complex (RISC), which is crucial for tissue development, stress physiology and activating immunity (*Chen, 2009*; *Huang et al., 2016*; *Khraiwesh et al., 2012*). The fungal plant pathogen *Botrytis cinerea*, secretes sRNAs that hijack the plant AGO/RISC in Arabidopsis, and *B. cinerea* sRNAs induce host gene silencing to support virulence (*Weiberg et al., 2013*), a mechanism known as cross-kingdom RNA interference (ck-RNAi) (*Weiberg et al., 2015*). In fungal-plant interactions, ck-RNAi is bidirectional, as plant-originated sRNAs are secreted into fungal pathogens and trigger gene silencing of virulence genes (*Cai et al., 2018*; *Zhang et al., 2016*). It is currently not known, how important ck-RNAi is for pathogen virulence in general and whether other kingdoms of microbial pathogens, such as oomycetes, transfer sRNAs into hosts to support virulence.

Oomycetes comprise some of the most notorious plant pathogens and belong to the eukaryotic phylum stramenopiles, which diverged from animals, plants and fungi over 1.5 billion years ago (*Parfrey et al., 2011*). Here, we demonstrate that sRNAs of the downy mildew causing oomycete *Hyaloperonospora arabidopsidis* are associated with the host plant's *Arabidopsis thaliana* AGO1/

RISC and that these mobile oomycete sRNAs are crucial for virulence by silencing plant host defence genes.

## Results

### Oomycete sRNAs associate with the plant AGO1

We used the oomycete *Hyaloperonospora arabidopsidis* isolate Noco2 as an inoculum that is virulent on the host plant *A. thaliana* ecotype Col-0 (*Knoth et al., 2007*). We presumed that *H. arabidopsidis* can produce sRNAs, as sRNA biogenesis genes like *RNA-dependent RNA polymerases* (*RDR*s) and *Dicer-like* (*DCL*) were discovered in the genome (*Bollmann et al., 2016*). In order to identify oomycete sRNAs that were expressed during infection and might be transferred into plant cells, we performed two types of sRNA-seq experiments. First, we sequenced sRNAs isolated from total RNA extracts at 4 and 7 days post inoculation (dpi) together with mock-treated plants. Second, we sequenced sRNAs isolated from *At*AGO1 immunopurification (*At*AGO1-IP) samples to seek for translocated oomycete sRNAs. We chose *At*AGO1-IP for sequencing, because *At*AGO1 is constitutively expressed and forms the major RISC in Arabidopsis (*Vaucheret, 2008*), and sRNAs of fungal pathogens were previously found to be associated with *At*AGO1 during infection (*Wang et al., 2016*; *Weiberg et al., 2013*). An overview of *A. thaliana* and *H. arabidopsidis* sRNA (*Hpa*sRNA) read numbers identified in all sRNA-seq experiments is given in *Supplementary file 1*. Size profiles of *Hpa*sRNA reads in total sRNA samples depicted two major peaks of 21 nucleotides (nt) and 25 nt (*Figure 1a*), suggesting that at least two categories of sRNAs occurred in this oomycete species. Similar sRNA size profiles were previously reported for plant pathogenic *Phytophthora* species (*Fahlgren et al., 2013*; *Jia et al., 2017*). The identified *Hpa*sRNAs mapped in different amounts to distinct regions of a *H. arabidopsidis* reference genome including ribosomal RNA (rRNA), transfer RNA (tRNA), small nuclear/nucleolar RNA (snRNA/snoRNA), protein-coding messenger RNA (mRNA, cDNA) and non-annotated regions (*Figure 1—figure supplement 1a*). After filtering out rRNA, tRNA and snRNA/snoRNA reads, *Hpa*sRNAs mapping to protein-coding genes and non-annotated regions still displayed 21 nt as well as 25 nt size enrichment (*Figure 1—figure supplement 1b*) with 5' terminal uracil (U) enrichment (*Figure 1b*). We also identified *Hpa*sRNA reads in the *At*AGO1-IP sRNA-seq data providing evidence that *Hpa*sRNAs associated with this host AGO-RISC. The *At*AGO1-associated *Hpa*sRNAs revealed a strong enrichment for 21 nt reads with 5' terminal U preference (*Figure 1c*). *At*AGO1 is known to bind preferentially endogenous 21 nt sRNAs with 5' terminal U (*Mi et al., 2008*), and we confirmed such *At*AGO1-binding preference to endogenous Arabidopsis sRNAs in our dataset (*Figure 1—figure supplement 1c*). Therefore, we suspected that *Hpa*sRNAs bound to *At*AGO1 during infection might have the potential to silence plant genes. To follow this line, we focussed on 133 unique *Hpa*sRNA reads that were present in the sRNA-seq data of total RNAs from infected samples with read counts > 5 reads per million and in at least one read in the *At*AGO1-IP sRNA-seq dataset. Among those, 34 *Hpa*sRNAs were predicted to target as a minimum one *A. thaliana* mRNA with stringent cut-off criteria. Most of the *At*AGO1-bound *Hpa*sRNAs with predicted Arabidopsis target genes mapped to non-annotated, intergenic regions in the *H. arabidopsidis* genome (*Supplementary file 2*). These *Hpa*sRNAs were found to be enriched in *At*AGO1-IP data compared to *At*AGO2-IP in an additional comparative AGO-IP sRNA-seq experiment (*Supplementary file 2*).

### Two predicted Arabidopsis mRNAs targets of *Hpa*sRNAs are down-regulated upon infection

In the following assays to investigate the function of *Hpa*sRNAs in ck-RNAi, we chose the *At*AGO1-enriched sRNA candidates *Hpa*sRNA2 and *Hpa*sRNA90. These two *Hpa*sRNAs were predicted to target the Arabidopsis *WITH NO LYSINE (K) KINASE* (*AtWNK*)2 and the extracellular protease *APOPLASTIC, ENHANCED DISEASE SUSCEPTIBILITY1-DEPENDENT* (*AtAED*)3, respectively (*Supplementary file 2*). We focussed on these two *Hpa*sRNAs and target genes, because *AtWNK2* and *AtAED3* mRNA levels were lower in leaves infected with a virulent *H. arabidopsidis* strain compared to an avirulent in a previous RNA-seq study (*Asai et al., 2014*), suggesting a negative impact of *H. arabidopsidis* proliferation on target transcript accumulation. Further on, members of the WNK protein family as well as *At*AED3 have been previously linked to plant stress response and immunity,

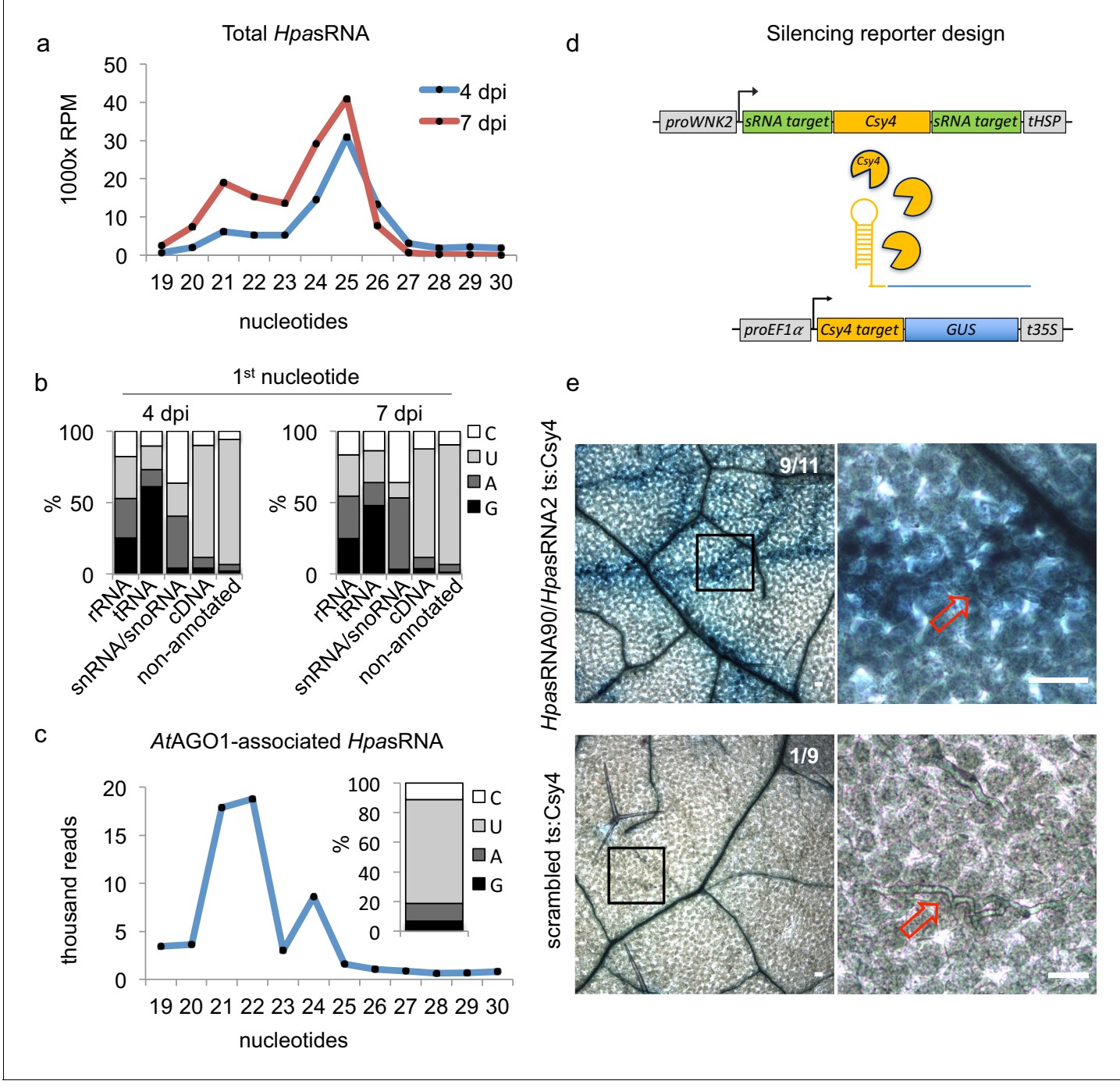

**Figure 1.** *Hpa*sRNAs translocated into the plant *At*AGO1 and induced host target silencing in infected plant cells. (**a**) Size profile of *Hpa*sRNAs revealed two size peaks at 21 nt and 25 nt at 4 and 7 dpi. (**b**) The frequency of the first nucleotide at 5' terminal positions of *Hpa*sRNAs mapping to cDNAs or non-annotated regions revealed bias towards uracil. (**c**) Size distribution and first nucleotide analysis of *At*AGO1-associated *Hpa*sRNAs showed size preference at 21 nt with 5' terminal uracil. (**d**) A novel Csy4/GUS reporter construct was assembled to detect *Hpa*sRNA-directed gene silencing, reporting GUS activity if *Hpa*sRNAs were functional to suppress Csy4 expression sequence-specificly. (**e**) GUS staining of infected leaves at two magnifications revealed sequence-specific reporter silencing at 4 dpi. Csy4 with *Hpa*sRNA2 and *Hpa*sRNA90 target sequences (ts) is depicted on the top and with random scrambled ts on the bottom. Red arrows indicate *H. arabidopsidis* hyphae in the higher magnification images. Scale bars indicate 50 μm. Numbers in the micrographs indicate number of leaves showing GUS activity per total leaves inspected.

The online version of this article includes the following figure supplement(s) for figure 1:

**Figure supplement 1.** Insights into the small RNAome of *H. arabidopsidis* and Arabidopsis.

**Figure supplement 2.** Stem-loop RT-PCR revealed *Hpa*sRNA2, *Hpa*sRNA30 and *Hpa*sRNA90 expression at 4 and 7 dpi in three biological replicates.

*Figure 1 continued on next page*

*Figure 1 continued*

**Figure supplement 3.** Relative expression of *AtAED3* and *AtWNK2* was measured in mock-treated or *H. arabidopsidis* inoculated plants.

**Figure supplement 4.** 5' RACE PCR did not provide evidence for pathogen sRNA mediated target cleavage.

**Figure supplement 5.** The reporter was neither activated by an endogenous miRNA target site nor by a distinct pathogen.

respectively (*Balakireva and Zamyatnin, 2018*; *Cao-Pham et al., 2018*). We confirmed expression of *Hpas*RNA2 and *Hpas*RNA90 in infected plants at 4 and 7 dpi by stem-loop reverse transcriptase (RT)-PCR (*Figure 1—figure supplement 2*). We then performed quantitative (q)RT-PCR to measure *AtWNK2* and *AtAED3* mRNAs expressed in whole seedling leaves of wild type (WT) plants upon *H. arabidopsidis* infection or mock treatment. We used the *atago1-27* mutant as a control line, because we anticipated that target suppression should fail in this mutant. Indeed, *AtAED3* was significantly down-regulated upon *H. arabidopsidis* inoculation at 7 dpi, and *AtWNK2* expression indicated moderate suppression at 4 dpi in WT plants, when compared to mock-treated plants (*Figure 1—figure supplement 3a*). Because the down-regulation effects were rather moderate, we repeated this experiment with a second independent *H. arabidopsidis* inoculation that validated the qRT-PCR results (*Figure 1—figure supplement 3b*). In support of *At*AGO1-mediated target silencing through *Hpas*RNAs, WT-like suppression of *AtWNK2* and *AtAED3* was not observed in the *atago1-27* background (*Figure 1—figure supplement 3*). However, *AtAED3* expression data also indicated down-regulation upon mock treatment during the course of the experiment that might have been caused by the almost 100% relative air humidity during the assay. Moreover, higher transcript levels were measured in *atago1-27* before infection when compared to WT plants.

As Arabidopsis target transcripts displayed expressional down-regulation upon *H. arabidopsidis* infection in WT plants, we wanted to explore, if *Hpas*RNAs guided mRNA slicing of *AtWNK2* and *AtAED3* through the host *At*AGO1/RISC during infection. *At*AGO1 possesses RNA cleavage activity on *At*miRNA-guided target mRNAs at the position 10/11 counted from the 5' end of the miRNA (*Mallory and Bouché, 2008*). We performed 5' rapid amplification of cDNA-ends (RACE)-PCR analysis to determine the 5' ends of target transcripts in RNAs isolated from infected plants pooled from 4 and 7 dpi. We isolated PCR products at the predicted cleavage sizes (*Figure 1—figure supplement 4a*) for next generation sequencing analysis. In total, we obtained 58,954 and 88,697 reads mapping to *AtWNK2* and *AtAED3*, respectively. However, only a small fraction of reads (639 for *AtWNK2* and 17 for *AtAED3*) mapped at the predicted target sites, while most reads aligned to further 3' downstream regions indicating rapid RNA degradation (*Figure 1—figure supplement 4b*). The 5' ends that matched to the predicted target sites did not display any predominant peak at the expected cleavage position 10/11, but were rather scattered over the entire target sites (*Figure 1—figure supplement 4c*). Therefore, RACE-PCR did not support *Hpas*RNA-guided cleavage of the Arabidopsis target mRNAs.

## *Hpas*RNAs translocate into Arabidopsis and induce host gene silencing in infected plant cells

To further examine if translocation of *Hpas*RNAs into Arabidopsis was sufficient to induce plant gene silencing during infection, we designed a novel *in situ* silencing reporter. This reporter is based on the CRISPR endonuclease Csy4 that specifically binds to and cleaves a short RNA sequence motif (*Haurwitz et al., 2010*). We fused this cleavage motif to a *β-glucuronidase* (*GUS*) reporter gene to mark it for degradation by Csy4 (*Figure 1d*). Further on, we cloned the native *AtWNK2* and *AtAED3* target sequences of *Hpas*RNA2 and *Hpas*RNA90 as flanking tags to the Csy4 coding sequence that turned Csy4 into a target of these *Hpas*RNAs. If *Hpas*RNAs would be capable of silencing effectively the Csy4 transgene, we expected an activation of GUS. Moreover, we constructed control reporters with either a scrambled target sequence or with the binding sequence taken from the endogenous *At*miRNA164 target gene *AtCUC2* (*Nikovics et al., 2006*) instead of the *Hpas*RNA2/*Hpas*RNA90 target sequences. With these control reporters, we intended to test if any *Hpas*RNA2/*Hpas*RNA90-independent suppression of Csy4 or any ck-RNAi-unrelated effect could result in GUS activation. Using the *At*miR164 target site, we anticipated to induce infection-independent local Csy4 silencing, because *At*miR164 expression in young, developing leaves was previously described to be locally restricted to defined regions at the leaf teeth and in the apical meristem (*Nikovics et al., 2006*). To

simulate *AtWNK2* target mRNA expression level of the Csy4 reporter transgene, we used a 2 kb-DNA fragment upstream of the *AtWNK2* start codon as a promoter sequence for all reporter constructs.

We transformed the reporter variants into Arabidopsis to examine the silencing efficiency of *Hpa*sRNAs on predicted plant targets upon infection. In each experiment, we tested at least three individual T1 lines per construct, and all plants appeared to be fully compatible with *H. arabidopsidis*. Csy4 successfully blocked GUS activity in plant cells that were not close to *H. arabidopsidis* infection sites (*Figure 1e*), providing evidence for functional *GUS* repression by Csy4. Plants expressing Csy4 transcripts fused to *Hpa*sRNA2 and *Hpa*sRNA90 target sequences highlighted GUS activation along the *H. arabidopsidis* hyphal infection front (*Figure 1e*). This experiment provided visual insights into the effective plant gene silencing by pathogen sRNAs, and thus let us assume that efficient sRNA translocation from the pathogen into the host cell occurred. GUS activity emerged only around the pathogen hyphae indicating that ck-RNAi did not spread further into distal regions away from primary infection sites. In contrast, Csy4 linked to a randomly scrambled or *At*miRNA164 target sequence did not express GUS activation around the *H. arabidopsidis* hyphae (*Figure 1e*, *Figure 1—figure supplement 5a*). We concluded that GUS activity induced by *H. arabidopsidis* in plants expressing Csy4 fused to *Hpa*sRNA2/*Hpa*sRNA90 target sites was neither due to target sequence-unspecific regulation of Csy4 or GUS nor due to pathogen-triggered regulation of the *AtWNK2* promoter. Moreover, reporter plants did also not display any local GUS activity at infection sites when inoculated with the unrelated oomycete pathogen *Phytophthora capsici* (*Figure 1—figure supplement 5b*). This result further supported that the GUS reporter was activated specifically by *Hpa*sRNAs and not by infection stress.

## Arabidopsis *atago1* exhibited enhanced disease resistance against downy mildew

Over one hundred *Hpa*sRNAs were detected to associate with the plant AGO1/RISC during infection, with 34 *Hpa*sRNAs being predicted to silence 49 plant targets including stress-related genes (*Supplementary file 2*). Such *Hpa*sRNAs can induce host target gene silencing at the infection site (*Figure 1e*). Based on these observations, we hypothesized that *At*AGO1 was relevant for *H. arabidopsidis* to suppress plant defence genes for infection. To test this hypothesis, we compared the disease outcome of *atago1-27* with WT plants. The *atago1-27* line represents a hypomorphic mutant, and developmental alterations are relatively mild compared to other *atago1* mutant alleles (*Morel et al., 2002*). Therefore, this *atago1* mutant line was suitable to perform pathogen infection assays. We stained infected leaves with Trypan Blue that visualized *H. arabidopsidis* infection structures and indicated plant cell death using a bright-field light microscope. The *atago1-27* plants exhibited a remarkable change of the disease phenotype by exhibiting dark Trypan Blue-stained host cells around hyphae instead of unstained plant cells colonized with *H. arabidopsidis* haustoria in WT plants (*Figure 2a*). We interpreted this disease phenotype in *atago1-27* plants as trailing necrosis of plant cells, which has been described for sub-compatible *A. thaliana*/*H. arabidopsidis* interactions (*Coates and Beynon, 2010*). Indeed, the trailing necrosis co-occurred with enhanced disease resistance, because *H. arabidopsidis* DNA content was strongly reduced (*Figure 2b*) and the number of *H. arabidopsidis* conidiospores was significantly lower in *atago1-27* (*Figure 2c*). Pathogen DNA content was also reduced in *atago1-27* cotyledons (*Figure 2—figure supplement 1a*) without displaying the trailing necrosis (*Figure 2—figure supplement 1b*). This reduced disease phenotype was linked to *atago1* mutations, as independent hypomorphic mutant alleles of *atago1-45* and *atago1-46* also displayed trailing necrosis after *H. arabidopsidis* inoculation, albeit to a smaller extent (*Figure 2—figure supplement 1c*). On the contrary, *atago2-1* and *atago4-2* did neither exhibit trailing necrosis nor reduced oomycete biomass (*Figure 2—figure supplement 1d–e*). We confirmed that *Hpa*sRNA2 and *Hpa*sRNA90 preferably bound to *At*AGO1 compared to *At*AGO2 by *At*AGO-IP coupled to stem-loop RT-PCR (*Figure 2—figure supplement 2*). This result was consistent with the observed reduced disease level in the *atago1* mutant lines in contrast to *atago2-1*.

Taken together, these data strongly suggested that translocated *Hpa*sRNAs act mainly through *At*AGO1 to suppress plant genes for infection. Nevertheless, increased disease resistance of *atago1* plants could have been caused by impaired function of plant endogenous sRNAs. For instance, *atago1* mutant plants as well as other miRNA pathway mutants, such as *atdcl1*, *athua enhancer(hen) 1 athasty(hst)* or *atserrate(se)* show pleiotropic developmental defects because of impaired plant

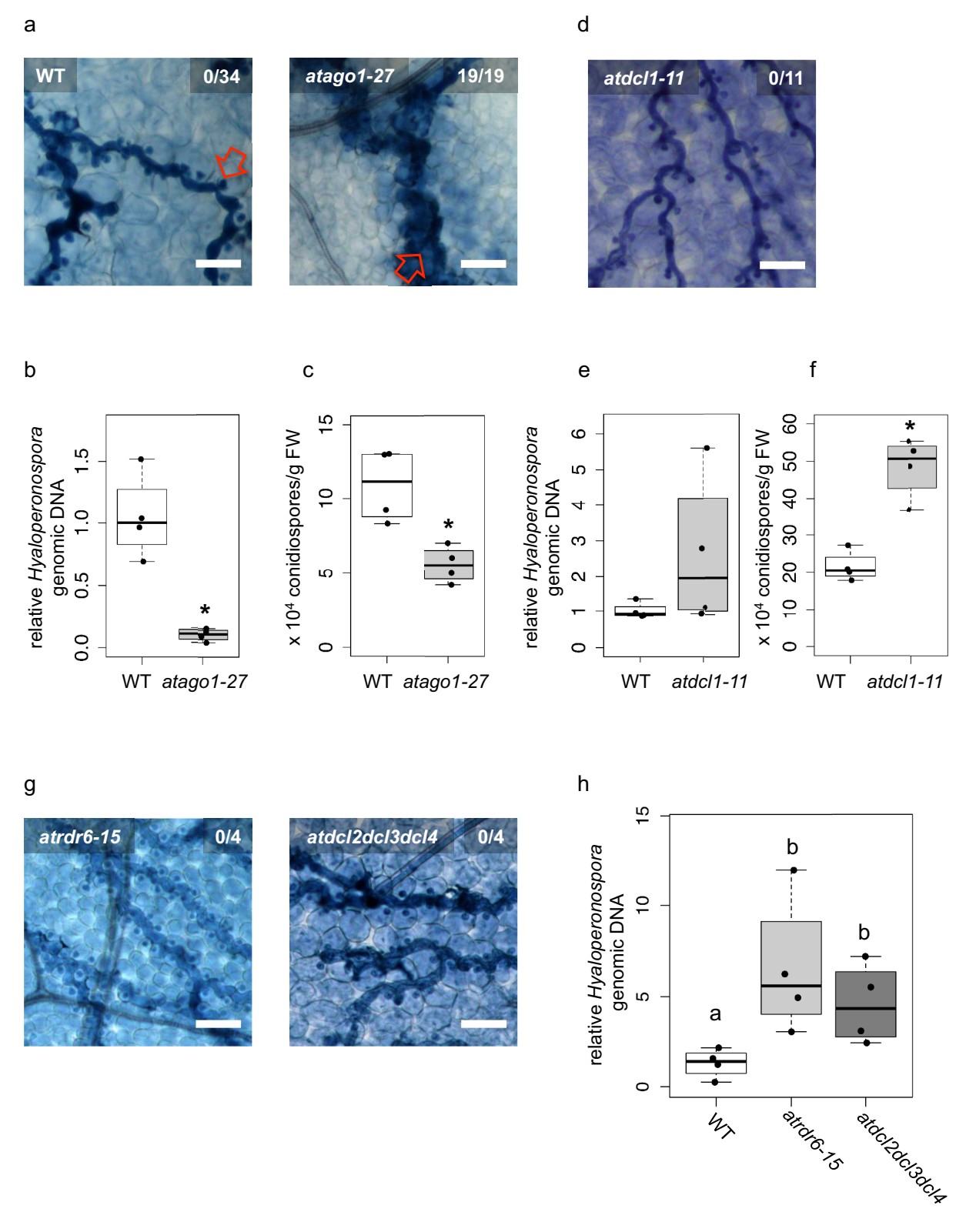

**Figure 2.** Arabidopsis *atago1* exhibited enhanced disease resistance against *H. arabidopsidis.* (a) Trypan Blue-stained microscopy images showed trailing necrosis around hyphae in *atago1-27*, but no necrosis on WT seedling leaves at 7 dpi. Red arrow in WT marks *H. arabidopsidis* haustorium, red arrow in *atago1-27* indicates trailing necrosis. (b) *H. arabidopsidis* genomic DNA was quantified in *atago1-27* and WT plants by qPCR at 4 dpi relative to plant genomic DNA represented by n ≥ four biological replicates. (c) Numbers of conidiospores per gram leaf fresh weight (FW) in *atago1-27* and

*Figure 2 continued on next page*

*Figure 2 continued*

WT plants at 7 dpi are represented by four biological replicates. (**d**) Trypan Blue-stained microscopy images of *atdcl1-11* did not show any trailing necrosis at 7 dpi. (**e**) *H. arabidopsidis* genomic DNA in *atdcl1-11* and WT plants at 4 dpi were in tendency enhanced with n ≥ four biological replicates. (**f**) Number of conidiospores per gram leaf fresh weight (FW) in *atdcl1-11* at 7 dpi was significantly elevated compared to WT plants. (**g**) Trypan Blue-stained microscopy images of *atrdr6-15* and *atdcl2dcl3dcl4* showed no plant cell necrosis after inoculation with *H. arabidopsidis* at 7 dpi. (**h**) *H. arabidopsidis* genomic DNA content in leaves was elevated in *atrdr6-15* and *atdcl2dcl3dcl4* compared to WT at 4 dpi with n ≥ four biological replicates. Asterisk indicates statistically significant difference by one tailed Student's t-test with p≤0.05. Letters indicate groups of statistically significant difference by ANOVA followed by TukeyHSD with p≤0.05. Scale bars in all microscopy images indicate 50 µm and numbers in the micrographs represent observed leaves with necrosis per total inspected leaves.

The online version of this article includes the following figure supplement(s) for figure 2:

**Figure supplement 1.** Enhanced resistance against infection was restricted to *atago1* mutants.

**Figure supplement 2.** Stem-loop RT-PCR of *Hpa*sRNAs from *At*AGO1-IP or *At*AGO2-IP of mock-treated or *H. arabidopsidis* infected leaf tissue.

**Figure supplement 3.** Trypan Blue-stained microscopy images presenting the *At*miRNA biogenesis mutants *athst-6*, *athen1-5* and *atse-2* did not show any trailing necrosis at 7 dpi.

**Figure supplement 4.** Common defence-related marker gene induction was not enhanced in *atago1-27* mutants.

**Figure supplement 5.** Relative mRNA expression of *AtRBOHD* and *AtRBOHF* determined by qRT-PCR using *AtActin* as reference in WT and *atago1-27* in *H. arabidopsidis* and mock treated plants.

**Figure supplement 6.** Susceptibility of *atago1* mutants to infection with the biotrophic fungus *E. cruciferarum* and the oomycete *A. laibachii* remained unaltered.

sRNA function (*Li and Zhang, 2016*; *Vaucheret, 2008*). To test whether other miRNA pathway mutants also revealed enhanced disease resistance similar to *atago1* plants, we inoculated the *atdcl1-11* mutant line with *H. arabidopsidis*. We did not detect any trailing necrosis or reduced pathogen biomass, but in contrast a significantly increased number of conidiospores (*Figure 2d–f*) indicating a positive role of *A. thaliana* miRNAs in immune response against *H. arabidopsidis*. These results provided evidence that necrotic trailing and reduced pathogen susceptibility found in *atago1* was not due to the loss of a functional plant miRNA pathway. In support, we did also not observe trailing necrosis upon infection in the *atse-2*, *athen1-5* and *athst-6* mutants (*Figure 2—figure supplement 3*).

Since *atago1* exhibited trailing necrosis and reduced susceptibility to *H. arabidopsidis,* we wanted to examine if constant activation of defence-related marker genes corresponded with enhanced disease resistance. We profiled gene expression of the *A. thaliana* immunity marker gene *AtPATHO-GENESIS-RELATED (PR)1*. *AtPR1* was neither faster nor stronger induced at 6, 12 or 18 h post inoculation in *atago1-27* compared to WT (*Figure 2—figure supplement 4a*). *AtPR1* and another immunity marker *AtPLANT-DEFENSIN (PDF)1.2* were not higher expressed in *atago1-27* at 1, 4 or 7 dpi compared to WT before or after infection (*Figure 2—figure supplement 4b–c*). To examine plant gene expression related to induced plant cell death, as observed in *ago1* mutants, we measured transcript levels of the two NADPH oxidases *At REACTIVE BURST OXIDASE HOMOLOG (AtR-BOH)D* and *AtRBOHF*. Both genes are required for accumulation of reactive oxygen intermediates to suppress spread of cell death during plant defence (*Torres et al., 2005*). Moreover, the *atrbohd* and *atrbohf* knockout mutant plants previously revealed increased plant cell death after *H. arabidopsidis* infection and were more resistant against this pathogen (*Torres et al., 2002*). In consistence, we found that *AtRBOHD* and *AtRBOHF* were induced in WT plants at 7 dpi and were significantly higher expressed than in *atago1-27* (*Figure 2—figure supplement 5*). These results gave a first hint of a host defence pathway that might be affected due to *At*AGO1-associated *Hpa*sRNAs.

Plant miRNAs can initiate the production of secondary phased siRNAs (phasiRNAs), which negatively control the expression of *NLR* (*NOD-like receptor*) class *Resistance* (*R*) genes (*Li et al., 2012*; *Shivaprasad et al., 2012*). Constitutive expression of *NLR* genes promotes immune responses such as spontaneous plant cell death resembling a hypersensitive response (*Lai and Eulgem, 2018*). Therefore, lack of phasiRNAs in *atago1* could cause enhanced expression of *NLR*s leading to resistance against *H. arabidopsidis*. To examine *R* gene-based enhanced resistance due to lack of phasiR-NAs, we inoculated the *atrdr6-15* and *atdcl2dcl3dcl4* mutants with *H. arabidopsidis* Noco2. The production of phasiRNAs depends on *At*RDR6 and *At*DCL2/*At*DCL3/*At*DCL4 (*Fei et al., 2013*). Both mutants did not exhibit trailing necrosis (*Figure 2g*), but in contrast highlighted increased pathogen biomass upon inoculation with *H. arabidopsidis* (*Figure 2h*). Higher susceptibility of *atrdr6-15* and

*atdcl2dcl3dcl4* to *H. arabidopsidis* was also in line with a previous report suggesting a role of Arabidopsis phasiRNAs in silencing of *Phytophothora* genes for host plant defence (*Hou et al., 2019*).

In order to further explore whether *atago1-27* was more resistant to other biotrophic fungi or oomycetes, we performed infection assays with the powdery mildew fungus *Erysiphe cruciferarum* and the white rust oomycete *Albugo laibachii*. We did not observe any plant cell necrosis in neither pathogen. Moreover, there was neither a reduction in the pustules for *A. laibachii* nor in pathogen biomass of *E. cruciferarum* (*Figure 2—figure supplement 6a–d*). Taken together, the observed disease resistance of *atago1* plants against *H. arabidopsidis* was probably neither based on increased basal plant immunity nor on *R* gene-mediated resistance.

## *Hpa*sRNAs are crucial for virulence

As we realized that *Hpa*sRNAs were associated with the host *At*AGO1-RISC, silenced plant target genes, and that Arabidopsis *atago1* mutants displayed reduced susceptibility towards *H. arabidopsidis* infection, we wanted to understand how important *Hpa*sRNAs were for *H. arabidopsidis* virulence. To shed light on the relevance of *Hpa*sRNAs for infection, we cloned and expressed a short-tandem-target-mimic (STTM) RNA in Arabidopsis to sequester *Hpa*sRNAs. The STTM strategy has been previously used to scavenge endogenous plant sRNAs and to prevent gene silencing of native target genes (*Tang et al., 2012*). We designed a triple STTM transgene to simultaneously bind the pathogen sRNAs *Hpa*sRNA2, *Hpa*sRNA30, and *Hpa*sRNA90 by RNA base-pairing. A non-complementary 3-base loop structure at the position 10/11 counted from the 5′ end of the *Hpa*sRNAs was deliberately incorporated to block potential cleavage by plant AGO/RISCs, as previously described (*Tang et al., 2012*; *Figure 3a*). We included the *At*AGO1-associated *Hpa*sRNA30 in the triple STTM, because it was predicted to silence *AtWNK5* (*Supplementary file 2*), a homolog of *AtWNK2*, thus we presumed that *Hpa*sRNA30-induced *AtWNK5* suppression might also be important for virulence. The *Hpa*sRNA30 sequence mapped only to the *H. arabidopsidis*, but not the Arabidopsis genome, and we detected this *Hpa*sRNA in infected plants at 4 and 7 dpi by sRNA-seq and stem-loop RT-PCR (*Figure 1—figure supplement 2*, *Supplementary file 2*). Remarkably, seven out of eleven individual STTM T1 transgenic lines resembled partially the trailing necrosis phenotype of *atago1* (*Figure 3b*). We isolated two stable STTM T2 lines (#4, #5). The STTM #4 line showed target de-repression of *AtAED3* at 7 dpi and of *AtWNK2* at 4 dpi upon *H. arabidopsidis* inoculation when compared to plants expressing an empty vector control (*Figure 3—figure supplement 1a*). These time points corresponded to target gene suppression as found by qRT-PCR analysis before (*Figure 1—figure supplement 3*). Moreover, both STTM T2 lines exhibited reduced pathogen biomass (*Figure 3—figure supplement 1b*) and allowed significantly lower production of pathogen conidiospores (*Figure 3c*). We also cloned STTMs against an rRNA-derived *Hpa*sRNA as well as against a random scrambled sequence for expression in Arabidopsis. These two types of control STTMs did not exhibit trailing necrosis in at least five independent T1 transgenic lines upon *H. arabidopsidis* inoculation (*Figure 3d*). Furthermore, we also did not observe disease resistance in transgenic plants expressing the STTM against *Hpa*sRNA2/*Hpa*sRNA30/*Hpa*sRNA90 when inoculated with the unrelated bacterial pathogen *Pseudomonas syringae* DC3000 (*Figure 3—figure supplement 1c*). These experiments provided evidence that the expression of anti-*Hpa*sRNA STTMs in Arabidopsis blocked *Hpa*sRNAs activity that resulted in reduced virulence of *H. arabidopsidis*.

## Arabidopsis target genes of *Hpa*sRNAs contribute to plant defence

Upon uncovering the importance of *Hpa*sRNAs for virulence, we wanted to assess the contribution of Arabidopsis target genes to plant defence. We obtained three T-DNA insertion lines for the identified target genes *AtWNK2* and *AtAED3*, namely *atwnk2-2*, *atwnk2-3*, and *ataed3-1* (*Figure 4—figure supplement 1a*). While *atwnk2-2* and *ataed3-1* are two SALK/SAIL lines (*Alonso et al., 2003*; *Sessions et al., 2002*) that carry a T-DNA insertion in their coding sequence, respectively, we now re-located the T-DNA insertion of the *atwnk2-3* plant line from the last exon into the 3′ UTR, based on sequencing the T-DNA flanking sites (*Figure 4—figure supplement 1a*). To study infection phenotypes, we stained *H. arabidopsidis*-infected leaves with Trypan Blue, and all T-DNA insertion lines resembled pathogen infection structures like in WT plants. However, haustorial density, indicated by the number of haustoria formed per intercellular hyphal distance, was significantly increased in *atwnk2-2* (*Figure 4—figure supplement 1b*). Intensified haustoria formation was previously

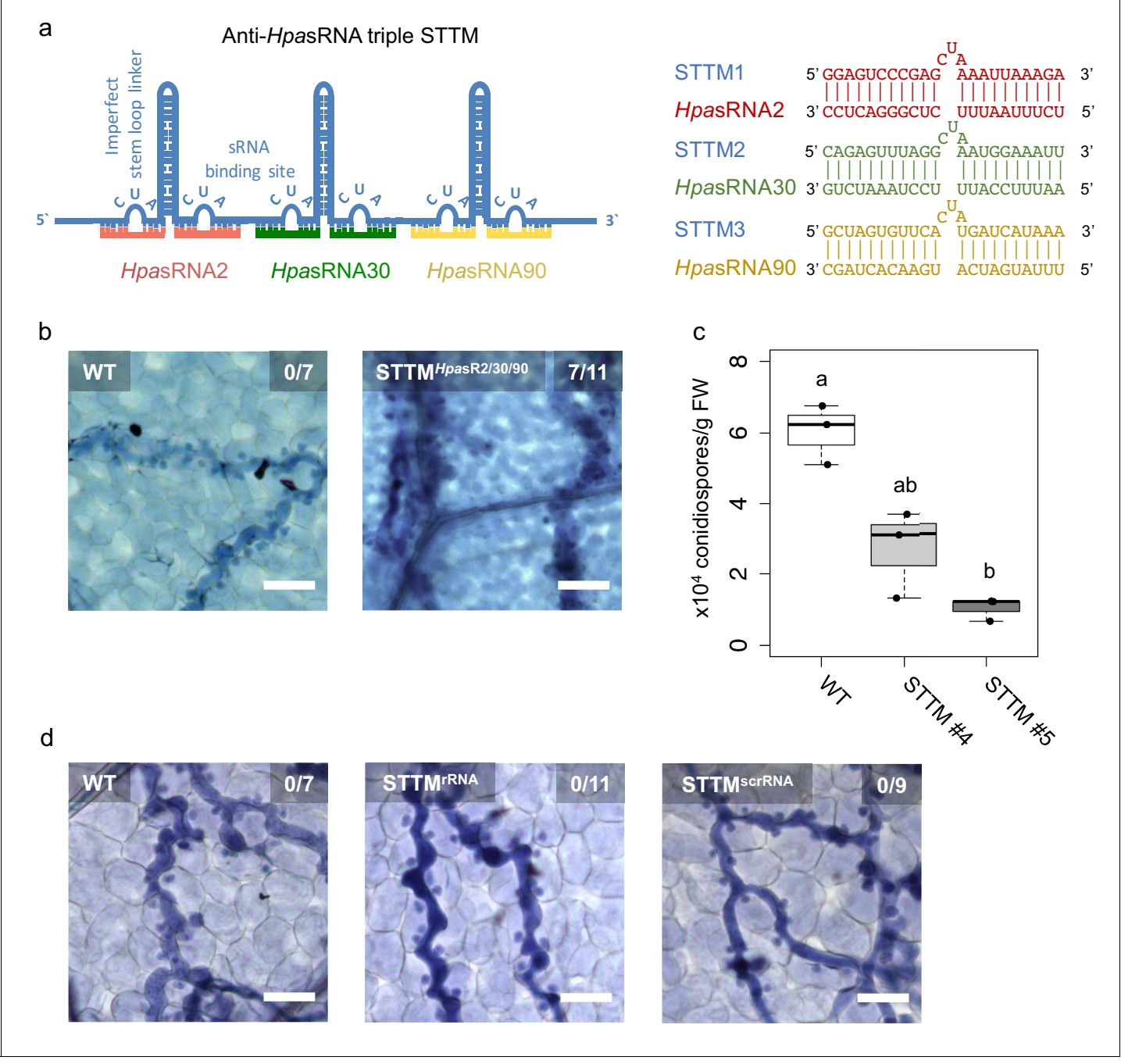

**Figure 3.** Translocated *Hpa*sRNAs were crucial for virulence. (a) A triple STTM construct was designed to target the three *Hpa*sRNAs *Hpa*sRNA2, *Hpa*sRNA30 and *Hpa*sRNA90 in Arabidopsis. (b) *A. thaliana* T1 plants expressing the triple STTM to scavenge *Hpa*sRNA2, *Hpa*sRNA30 and *Hpa*sRNA90 exhibited trailing necrosis at 7 dpi. (c) Number of conidiospores per gram FW was significantly reduced in two independent STTM-expressing Arabidopsis T2 lines (#4, #5) compared to WT. (d) Transgenic Arabidopsis plants in T1 expressing a STTM complementary to a rRNA-derived *Hpa*sRNA (STTM^rRNA) or to a random scrambled (STTM^scrRNA) sequence did not exhibit trailing necrosis at 7 dpi. The scale bars indicate 50 μm and numbers represent observed leaves with necrosis per total inspected leaves.

The online version of this article includes the following figure supplement(s) for figure 3:

**Figure supplement 1.** STTM plants revealed higher expression of target genes and lower *H. arabidopsidis* abundance.

interpreted as a sign of enhanced susceptibility in other plant/downy mildew pathogen interactions (*Hooftman et al., 2007*; *Unger et al., 2007*). Moreover, the pathogen DNA content was slightly but not significantly increased in *atwnk2-2* and *ataed3-1* compared to WT plants, but this was not the case for *atwnk2-3* (*Figure 4a*). Nevertheless, a significantly increased number of conidiospores (*Figure 4b*) and sporangiophores (*Figure 4c*) was observed in all the tested *atwnk2* and *ataed3* mutant lines upon *H. arabidopsidis* infection compared to WT plants.

We wanted to investigate in more detail the effect of target gene silencing by *Hpa*sRNAs on plant defence. For this, we cloned *AtWNK2* and *AtAED3* target genes either as native versions or artificially introduced synonymous point mutations in the target sites of *Hpa*sRNAs to generate the target gene-resistant versions *AtAED3r* and *AtWNK2r* (*Figure 4—figure supplement 2*). We transformed these gene versions into the respective mutant background *ataed3-1* and *atwnk2-2* expressing them under the control of their native promoters. Transgenic *AtWNK2* and *AtWNK2r* expressing plants reverted from previously described early flowering of *atwnk2-2* (*Wang et al., 2008*) into the WT phenotype validating successful complementation of *atwnk2-2* (*Figure 4—figure supplement 3*). If *AtWNK2* and *AtAED3* silencing through *Hpa*sRNA2 or *Hpa*sRNA90 was relevant to plant defence, we would expect that *AtWNK2r* and *AtAED3r* expressing plants become more resistant against *H. arabidopsidis*. Both, the native gene versions and the target site resistant versions, exhibited reduced number of conidiospores compared to T-DNA mutant plants transformed with an empty expression vector, respectively (*Figure 4d*). To further explore the role of target genes in plant immunity, we attempted to generate overexpression lines of resistant target gene versions by using the strong *Lotus japonicus Ubiquitin1* promoter (*proLjUbi1*) (*Maekawa et al., 2008*). We obtained an overexpressor line of the *AtWNK2r* version (*AtWNK2r-OE*) in the *atwnk2-2* background. These *AtWNK2r-OE* plants showed ectopic cell death in distance from infection sites (*Figure 4—figure supplement 4a*), as previously described for overexpression lines of other immunity factors, such as *AtBAK1* (*Domínguez-Ferreras et al., 2015*). Moreover, infection structures frequently displayed aberrant swelling-like structures and extensive branching of hyphae instead of the regular pyriform haustoria formed in *atwnk-2–2* (*Figure 4—figure supplement 4b*), further indicating a role for *AtWNK2* in immune reaction.

To gain more information on the conservation of the 34 identified *At*AGO1-associated *Hpa*sRNAs (*Supplementary file 2*), we analysed RNA sequence diversity using the *H. arabidopsidis* sequenced genomes of the Noco2, Cala2 and Emoy2 isolates (NCBI BioProject IDs: PRJNA298674; PRJNA297499, PRJNA30969). In a complementary approach, we investigated the variation of the 49 predicted plant target sites among 1135 *A. thaliana* genome sequenced accessions published by the 1001 genome project (*1001 Genomes Consortium, 2016*). Interestingly, all *HpasRNA* were found by BLASTn search in the three *H. arabidopsidis* isolates with only three allelic variations identified in Emoy2 (*Figure 4—figure supplement 5a*). On the Arabidopsis target site, we found single nucleotide polymorphisms (SNPs) and indels in 70% of all target genes (*Supplementary file 2*), many of those might impair in the predicted *Hpa*sRNA-induced silencing (*Figure 4—figure supplement 5b*). Of note, the *HpasRNA2* sequence was deeper conserved in other pathogenic oomycete species, compared to other *Hpa*sRNAs described in this study (*Figure 4—figure supplement 6a*). Moreover, the predicted target sites of the pathogen *siR2* homologs lie within a conserved region of other plant *WNK2* orthologs, with the lowest number of base pair mismatches occurring in the highly-adapted *A. thaliana*/*H. arabidopsidis* interaction (*Figure 4—figure supplement 6b*). Whether RNA sequence diversity in *Hpa*sRNAs and *A. thaliana* target mRNAs drives co-evolution in this co-adapted plant-pathogen system, remains to be further investigated.

## Discussion

In this study, we discovered that ck-RNAi happened during *H. arabidopsidis* host infection and contributed to the virulence of this pathogen. Sequencing sRNAs associated with Arabidopsis AGO1 revealed at least 34 *Hpa*sRNAs that entered the host RNAi machinery and potentially targeted multiple plant genes for silencing. These deep sequencing data offered first insights into the *H. arabidopsidis* sRNA transcriptome during host infection. Total read numbers of *At*AGO1-bound *Hpa*sRNAs were in the ratio of around 1/1000 compared to *At*AGO1-bound Arabidopsis sRNAs, raising the concern that concentration of pathogen sRNAs might not be sufficient to be functional. Nevertheless, our and other studies found genetic and phenotypic evidence for pathogen oomycete sRNA

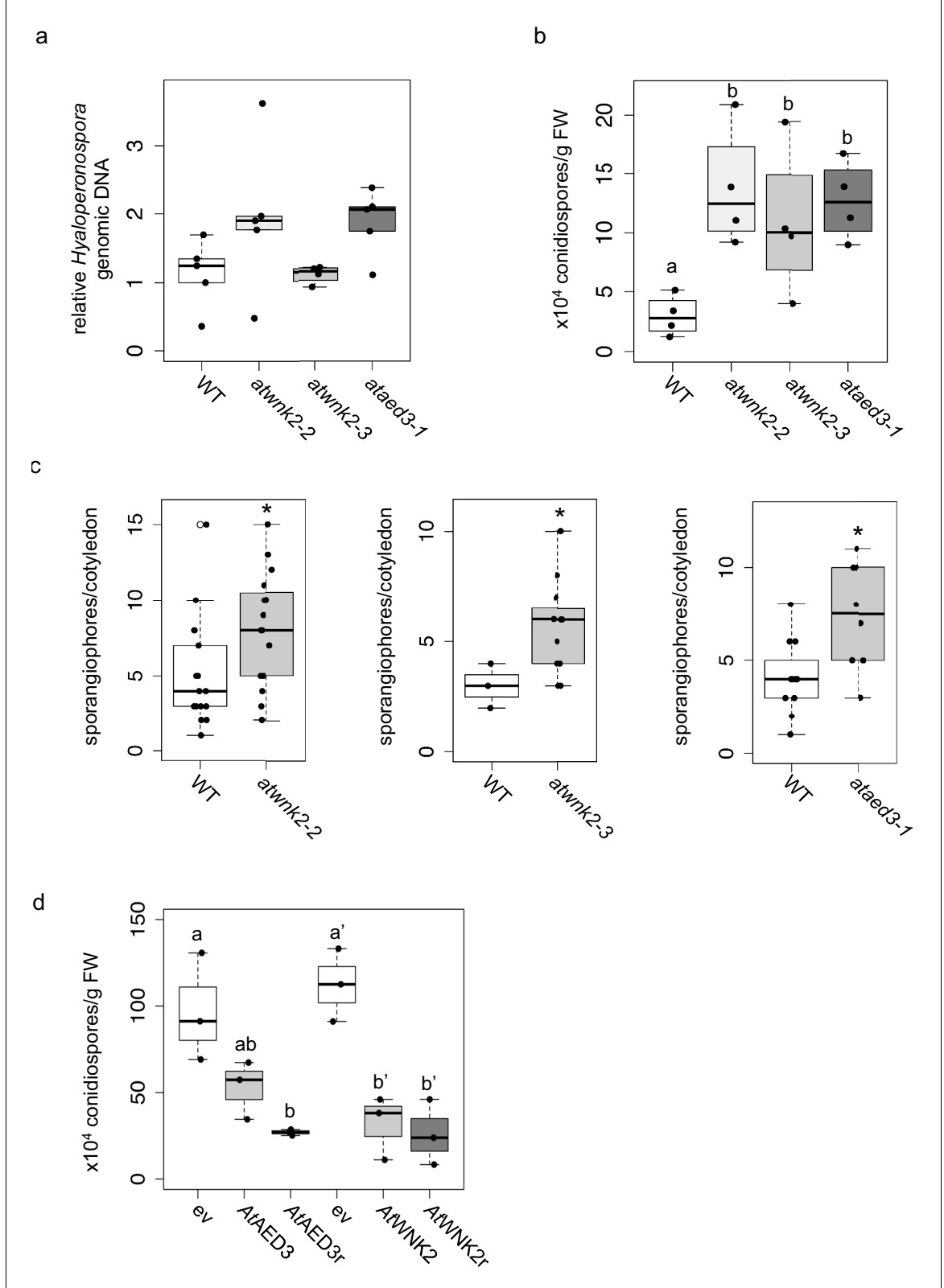

**Figure 4.** Arabidopsis target genes of *Hpa*sRNAs contributed to plant defence. (a) *H. arabidopsidis* genomic DNA content in leaves was slightly but not significantly enhanced in *atwnk2-2* and *ataed3-1* compared to WT, but not in *atwnk2-3*, at 4 dpi with n ≥ four biological replicates. (b) T-DNA insertion lines of *Hpa*sRNA target genes *ataed3-1*, *atwnk2-2*, and *atwnk2-3* showed significantly higher number of sporangiophores per cotyledon upon infection compared to WT at 5 dpi. (c) *ataed3-1*, *atwnk2-2*, and *atwnk2-3* showed significantly higher numbers of conidiospores per gram leaf FW upon

*Figure 4 continued on next page*

Figure 4 continued

infection compared to WT at 5 dpi. (d) Number of conidiospores was significantly reduced in gene-complemented mutant lines using the corresponding native promoters proAtEWNK2 or proAtAED3 with native gene sequence, AtAED3 and AtWNK2, or with target site resistant versions, AtAED3r and AtWNK2r compared to the knockout mutant background expressing an empty vector (ev), respectively. Asterisks indicate significant difference by one tailed Student's t-test with p≤0.05. Letters indicate significant difference by one-site ANOVA test.

The online version of this article includes the following figure supplement(s) for figure 4:

**Figure supplement 1.** Further details on sRNA target gene mutants.

**Figure supplement 2.** Target sequence-resistant versions of AtAED3 (AtAED3r) and AtWNK2 (AtWNK2r) were created by introducing synonymous nucleotide substitutions indicated by red letters.

**Figure supplement 3.** Transgenic A. thaliana atwnk2-2 was complemented with proWNK2:WNK2 or proWNK2:WNK2r that resulted in a WT-like flowering time point, while empty vector (ev) exhibited early flowing phenotype, as reported for atwnk2-2 (**Wang et al., 2008**).

**Figure supplement 4.** A. thaliana plants overexpressing proLjUBI1:AtWNK2r in the atwnk2-2 background revealed local necrosis without pathogen infection (a) and aberrant hyphae and haustoria swellings (b).

**Figure supplement 5.** Sequence diversity of HpasRNAs and their predicted Arabidopsis target mRNAs.

**Figure supplement 6.** The pathogen sRNA2 and its target are conserved across different plant pathogenic oomycetes and hosts.

function despite read numbers being in the range of ten per million or lower (**Jahan et al., 2015**; **Qutob et al., 2013**). By designing a novel Csy4/GUS repressor reporter system, we demonstrated that HpasRNAs have the capacity to translocate into plant cells and suppress host target genes. This new reporter system was capable of visualizing local gene silencing alongside the H. arabidopsidis hyphae. Therefore, the relatively small proportion of HpasRNAs counted in AtAGO1 sRNA-seq experiment could be explained by strong dilution with AtAGO1 molecules purified from non-colonized tissue. For the same reason, we measured moderate AtWNK2 and AtAED3 target gene suppression due to dilution effects coming from non-infected leaf lamina.

We assumed that diverse HpasRNAs were translocated into Arabidopsis during infection and AtAGO1 was a major hub of HpasRNAs, as detected by AtAGO1 pull down and sRNA-seq analysis. By which pathways and mechanisms HpasRNAs move into plant cells remains an open question. Transport via the extrahaustorial matrix could be a realistic cross-point, as many other biomolecules are exchanged via this route from pathogen to plant cells and vice versa (**Judelson and Ah-Fong, 2019**). It is noteworthy that accumulation of vesicle-like structures was visualized by electron microscopy at the perihaustorial matrix (**Mims et al., 2004**). In this regard, transfer of plant sRNAs into pathogen cells via exosomal vesicles was reported to induce ck-RNAi (**Cai et al., 2018**; **Hou et al., 2019**), making extracellular vesicles a prime suspect for HpasRNA transport into plant cells.

Plant RISC-associated HpasRNAs were crucial for successful infection, because transgenic Arabidopsis generated to block the suppressive function of the three candidate HpasRNA2, HpasRNA30 and HpasRNA90 via STTM target mimics diminished H. arabidopsidis virulence. As we identified 34 AtAGO1-associated HpasRNAs with 49 predicted plant target genes, we suggest that many HpasRNAs collaboratively sabotage gene expression of the plant immune response. Such a collaborative function was also suggested for proteinaceous pathogen effectors (**Cunnac et al., 2011**).

Regarding the role of identified HpasRNA target genes in host defence, our data supported quantitative contributions of AtAED3 and AtWNK2 to plant immunity. AtAED3 encodes a putative apoplastic aspartyl protease and has been suggested to be involved in systemic immunity (**Breitenbach et al., 2014**). AtWNK2 contributes to flowering time regulation in A. thaliana, while other members of the plant WNK family have been linked to the abiotic stress response (**Cao-Pham et al., 2018**). What is the particular function of these target genes against H. arabidopsidis infection and whether these also play a role against other pathogens, still needs to be explored.

The fact that Arabidopsis siRNA biogenesis mutants like atrdr6-15 and atdcl2dcl3dcl4 displayed increased H. arabidopsidis growth is an indication for the important role of secondary phasiRNAs in plant immunity, that was already observed against fungal pathogens like Verticillium dahliae and Magnaporthe oryzae (**Ellendorff et al., 2009**; **Wagh et al., 2016**). This is likely due to the regulatory function of phasiRNAs on endogenous plant immunity genes including the NLRs (**Li et al., 2012**; **Shivaprasad et al., 2012**). Two recent studies suggested suppressive roles of secreted plant phasiRNAs in ck-RNAi by silencing fungal B. cinerea and oomycete P. capsici virulence genes (**Cai et al., 2018**; **Hou et al., 2019**). Interestingly, exogenously applied sRNAs targeting the Cellulose synthase 3A gene of H. arabidopsidis can lead to pathogen developmental changes and spore germination

inhibition, suggesting functional RNA uptake by this pathogen (*Bilir et al., 2019*). Together with our data, we think that ck-RNAi in *H. arabidopsidis*/Arabidopsis interaction is bidirectional, as already described in fungal-plant interactions (*Cai et al., 2018*; *Wang et al., 2016*).

This study provides evidence that ck-RNAi, originally discovered in the fungal plant pathogen *B. cinerea* (*Weiberg et al., 2013*), is part of virulence in the oomycete biotrophic pathogen *H. arabidopsidis*. The phenomenon of plant-pathogen ck-RNAi is further proposed in the cereal fungal pathogens *Puccinia striiformis* (*Wang et al., 2017*) and *Blumeria graminis* (*Kusch et al., 2018*). We did not notice any enhanced resistance in an Arabidopsis *atago1* mutant against the biotrophic fungus *E. cruciferarum* and the oomycete *A. laibachii*, making ck-RNAi via *At*AGO1 unlikely. Further experiments are needed to rule out any importance of ck-RNAi for virulence of these two pathogens via alternative plant AGO-RISCs. The fungal wheat pathogen *Zymoseptoria tritici* was reported to not induce ck-RNAi (*Kettles et al., 2019*; *Ma et al., 2020*), while the corn smut pathogen *Ustilago maydis* has lost its canonical RNAi machinery (*Kämper et al., 2006*; *Laurie et al., 2008*). It will be interesting to elucidate why some pathogens have evolved ck-RNAi, while some others not.

# Materials and methods

## Key resources table

| Reagent type (species) or resource | Designation | Source or reference | Identifiers | Additional information |
|---|---|---|---|---|
| Gene (*Arabidopsis thaliana*) | AtWNK2 | arabidopsis.org | AT3G22420 | |
| Gene (*Arabidopsis thaliana*) | AtAED3 | arabidopsis.org | AT1G09750 | |
| Gene (*Arabidopsis thaliana*) | AtPR1 | arabidopsis.org | AT2G14610 | |
| Gene (*Arabidopsis thaliana*) | AtPDF1.2 | arabidopsis.org | AT5G44420 | |
| Gene (*Arabidopsis thaliana*) | AtAGO1 | arabidopsis.org | AT1G48410 | |
| Gene (*Arabidopsis thaliana*) | AtAGO2 | arabidopsis.org | AT1G31280 | |
| Strain, strain background (*Hyaloperonospora arabidopsidis*) | Noco2 | isolated originally in Norwich, UK | | |
| Strain, strain background (*Albugo laibachii*) | Nc14 | *Kemen et al., 2011* DOI: 10.1371/journal.pbio.1001094 | | |
| Strain, strain background (*Pseudomonas syringae* pv *tomato*) | DC3000 | *Whalen et al., 1991* DOI: 10.1105/tpc.3.1.49 | | |
| Strain, strain background (*Phytophthora capsici*) | LT263 | *Hurtado-Gonzales and Lamour, 2009* DOI: 10.1111/j.1365–3059.2009.02059.x | | |
| Genetic reagent (*Arabidopsis thaliana*) | atago1-27 | *Morel et al., 2002* PMID:11910010 | | |
| Genetic reagent (*Arabidopsis thaliana*) | atago1-45 | Nottingham Arabidopsis stock center (NASC) | N67861 | |
| Genetic reagent (*Arabidopsis thaliana*) | atago1-46 | (Nottingham Arabidopsis stock center (NASC) | N67862 | |
| Genetic reagent (*Arabidopsis thaliana*) | atago2-1 | *Takeda et al., 2008* DOI: 10.1093/pcp/pcn043 | | |
| Genetic reagent (*Arabidopsis thaliana*) | atago4-2 | *Agorio and Vera, 2007* DOI: 10.1093/pcp/pcn043 | | |

*Continued on next page*

*Continued*

| Reagent type (species) or resource | Designation | Source or reference | Identifiers | Additional information |
|---|---|---|---|---|
| Genetic reagent (*Arabidopsis thaliana*) | *atdcl1-11* | *Zhang et al., 2008* DOI: 10.1111/j.1365–3040.2008.01786.x | | |
| Genetic reagent (*Arabidopsis thaliana*) | *atdcl2dcl3dcl4* | *Deleris et al., 2006* DOI: 10.1126/science.1128214 | | triple mutant |
| Genetic reagent (*Arabidopsis thaliana*) | *athen1-5* | *Vazquez et al., 2004* DOI: 10.1016/j.cub.2004.01.035 | | |
| Genetic reagent (*Arabidopsis thaliana*) | *athst-6* | *Bollman et al., 2003* PMID:12620976 | | |
| Genetic reagent (*Arabidopsis thaliana*) | *atrdr6-15* | *Allen et al., 2004* DOI: 10.1038/ng1478 | | |
| Genetic reagent (*Arabidopsis thaliana*) | *atse-2* | *Grigg et al., 2005* DOI: 10.1038/nature04052 | | |
| Genetic reagent (*Arabidopsis thaliana*) | *proAGO2:HA-AGO2* | *Montgomery et al., 2008* DOI: 10.1016/j.cell.2008.02.033 | | |
| Genetic reagent (*Arabidopsis thaliana*) | *atwnk2-2* (SALK_121042) | Nottingham Arabidopsis stock center (NASC) | N663846 | |
| Genetic reagent (*Arabidopsis thaliana*) | *atwnk2-3* (SALK_206118) | Nottingham Arabidopsis stock center (NASC) | N695550 | |
| Genetic reagent (*Arabidopsis thaliana*) | *ataed3-1* (SAIL_722_G02C1) | Nottingham Arabidopsis stock center (NASC) | N867202 | |
| Genetic reagent (*Arabidopsis thaliana*) | *proLjUBI:STTMHasR2: STTMHasR30:STTMHasR90* | this study | | stable triple STTM overexpressor line (maintained in the Weiberg lab) |
| Genetic reagent (*Arabidopsis thaliana*) | *proAtWNK2:HasRNA2/90ts: Csy4:HasRNA2/90ts; proEF1:Csy4ts:GUS* | this study | | stable silencing reporter line (maintained in the Weiberg lab) |
| Genetic reagent (*Arabidopsis thaliana*) | *proAtWNK2:AtmiR164ts: Csy4:AtmiR164ts; proEF1:Csy4ts:GUS* | this study | | stable silencing reporter line (maintained in the Weiberg lab) |
| Genetic reagent (*Arabidopsis thaliana*) | *proAtWNK2:scrambled: Csy4:scrambled; proEF1:Csy4ts:GUS* | this study | | stable silencing reporter line (maintained in the Weiberg lab) |
| Genetic reagent (*Arabidopsis thaliana*) | *atwnk2-2* (*proAtWNK2:AtWNK2-GFP*) | this study | | stable *WNK2* complementation line (maintained in the Weiberg lab) |
| Genetic reagent (*Arabidopsis thaliana*) | *atwnk2-2* (*proAtWNK2:AtWNK2r-GFP*) | this study | | stable, sRNA resistant *WNK2* complementation line (maintained in the Weiberg lab) |
| Genetic reagent (*Arabidopsis thaliana*) | *atwnk2-2* (*proAtWNK2:GFP*) | this study | | stable plant line as empty vector control (maintained in the Weiberg lab) |
| Genetic reagent (*Arabidopsis thaliana*) | *ataed3-1* (*proAtAED3:AtAED3-GFP*) | this study | | stable *AED3* complementation line (maintained in the Weiberg lab) |
| Genetic reagent (*Arabidopsis thaliana*) | *ataed3-1* (*proAtAED3:AtAED3r-GFP*) | this study | | stable, sRNA resistant *AED3* complementation line (maintained in the Weiberg lab) |
| Genetic reagent (*Arabidopsis thaliana*) | *ataed3-1* (*proAtAED3: GFP*) | this study | | stable plant line as empty vector control (maintained in the Weiberg lab) |

*Continued on next page*

*Continued*

| Reagent type (species) or resource | Designation | Source or reference | Identifiers | Additional information |
|---|---|---|---|---|
| Antibody | anti-*At*AGO1 (rabbit polyclonal) | Agrisera | AS09 527; RRID:AB_2224930 | IP(1 μg antibody/g tissue), WB (1:4000) |
| Antibody | anti-HA (3F10; rat monoclonal) | Roche Diagnostics | Sigma-Aldrich (11867423001); RRID:AB_2314622 | IP(0.1 μg antibody/g tissue), WB (1:1000) |
| Antibody | anti-HA (12CA5; mouse monoclonal) | provided by Dr. Michael Boshart | | IP(0.1 μg antibody/g tissue), WB (1:1000), available in the Boshart lab (LMU Munich) |
| Antibody | anti-mouse IRdye800 (goat polyclonal) | Li-Cor | 926–32210; RRID:AB_2782998 | secondary antibody WB (1:15000) |
| Antibody | anti-rat IRdye800 (goat polyclonal) | Li-Cor | 926–32219; RRID:AB_1850025 | secondary antibody WB (1:15000) |
| Antibody | anti-rabbit IRdye800 (goat polyclonal) | Li-Cor | 926–32211; RRID:AB_621843 | secondary antibody WB (1:3000) |
| Commercial assay or kit | NEBNext Multiplex Small RNA Library Prep Set for Illumina | New England Biolabs (NEB) | NEB: E7300 | |
| Commercial assay or kit | 5′/3′ RACE Kit, 2nd Generation | Roche Diagnostics | Sigma-Aldrich: 03353621001 | |
| Commercial assay or kit | sparQ DNA Library Prep Kit | Quantabio | vwr.com (95191–024) | |
| Software, algorithm | Galaxy Server | *Giardine et al., 2005* | | hosted by the Gene Center Munich |

## Plant material

*Arabidopsis thaliana* (L.) seedlings were grown on soil under long day conditions (16 hr light/8 hr dark, 22°C, 60% relative humidity). The *atago1-27, atago1-45, atago1-46, atago2-1, atago4-2, athst-6, athen1-5, atse-2, atdcl1-11, atdcl2dcl3dcl4, atrdr6-15,* and *proAGO2:HA-AGO2* mutant lines (all in the Col-0 background) were described previously (*Agorio and Vera, 2007*; *Allen et al., 2004*; *Bollman et al., 2003*; *Deleris et al., 2006*; *Grigg et al., 2005*; *Morel et al., 2002*; *Smith et al., 2009*; *Takeda et al., 2008*; *Vazquez et al., 2004*; *Zhang et al., 2008*; *Montgomery et al., 2008*). The *atwnk2-2* (SALK_121042, [*Wang et al., 2008*]), *atwnk2-3* (SALK_206118) and *ataed3-1* (SAIL_722_G02C1) lines were verified for the T-DNA insertion by PCR on genomic DNA.

## *Hyaloperonospora arabidopsidis* inoculation

*Hyaloperonospora arabidopsidis* (Gäum.) isolate Noco2 was maintained on Col-0 plants. Plant inoculation was performed using $2\text{–}2.5 \times 10^4$ spores/ml and inoculated plants were incubated as described previously (*Ried et al., 2019*). For *atwnk2-2, atwnk2-3, and ataed3-1* pathogen assays inoculum strength was reduced to $1 \times 10^4$ spores/ml.

## *Albugo laibachii* (Thines and Y.J. Choi) inoculation

Plants were grown in short-day conditions (10 hr light, 22°C, 65% humidity/14 hr dark, 16°C, 60% humidity, photon flux density 40 μmol $m^{-2}$ $s^{-1}$) and inoculated at the age of six weeks. *A. laibachii* (isolate Nc14; [*Kemen et al., 2011*]) zoospores obtained from propagation on Arabidopsis accession Ws-0 were suspended in water ($10^5$ spores $ml^{-1}$) and incubated on ice for 30 min. The spore suspension was filtered through Miracloth (Calbiochem, San Diego, CA, USA) and sprayed onto the plants using a spray gun (~700 μl/plant). Plants were incubated at 8°C in a cold room in the dark overnight. Inoculated plants were kept under 10 hr light/14 hr dark cycles with a 20 °C day and 16°C night temperature. Infection rates were determined at 21 dpi for 12 individuals per WT and mutants by visual infection intensity.

## Powdery mildew inoculation

*Erysiphe cruciferarum* (Opiz ex L. Junell) was maintained on highly susceptible Col-0 *phytoalexin deficient* (*pad*)*4* mutants in a growth chamber at 22°C, a 10 hr photoperiod with 150 µmol m$^{-2}$s$^{-1}$, and 60% relative humidity. For pathogen assays 6 week-old Arabidopsis plants were inoculated with *E. cruciferarum* in a density of 3–5 spores mm$^{-2}$ and replaced under the same conditions.

## *Pseudomonas* pathogen assay

*Pseudomonas syringae* pv. *tomato* DC3000 was streaked from a freezer stock onto LB agar plates with Rifampicin. A single colony was used for inoculation of an overnight culture in liquid LB with Rifampicin. *Pseudomonas* was resuspended in 10 mM MgCl$_2$ and bacteria concentration was adjusted to OD$_{600}$ = 0.0002. 5–6 week-old Arabidopsis grown under short day conditions were leaf infiltrated using a needleless syringe, dried for 2 h and incubated under long day conditions. At 3 dpi, three leaf discs per plant (Ø=0.6 cm) were harvested and homogenized in 10 mM MgCl$_2$ for one biological replicate. Bacteria populations were counted as colony forming units using a serial dilution spotted on LB agar plates with Rifampicin.

## *Phytophthora capsici* (Leonian) inoculation

*Phytophthora capsici* LT263 (*Hurtado-Gonzales and Lamour, 2009*) was maintained on rye agar plates (*Caten and Jinks, 1968*). Agar plugs from fresh mycelium (Ø=0.4 cm) were placed on leaves of 5–6 week-old Arabidopsis plants grown under short day conditions. After 24 hr, plugs were removed and leaves were taken for GUS staining at 48 and 72 hpi.

## Trypan Blue staining

Infected leaves were stained with Trypan Blue as described previously (*Koch and Slusarenko, 1990*). Microscopic images were taken with a DFC450 CCD-Camera (Leica) on a CTR 6000 microscope (Leica Microsystems).

## GUS staining

Infected leaves were vacuum-infiltrated with GUS staining solution (0.625 mg ml$^{-1}$ X-Gluc, 100 mM phosphate buffer pH 7.0, 5 mM EDTA pH 7.0, 0.5 mM K$_3$[Fe(CN)$_6$], 0.5 mM K$_4$[Fe(CN)$_6$], 0.1% Triton X-100) and incubated over night at 37°C. Leaves were de-stained with 70% ethanol overnight and microscopic images were taken with the same microscopy set up as Trypan Blue stained samples.

## Pathogen quantification

*H. arabidopsidis* spores were harvested at 7 dpi into 2 ml of water. The spore concentration was determined using a haemocytometer (Neubauer improved, Marienfeld). The sporangiophore number was counted on detached cotyledons using a binocular. For biomass estimation, genomic DNA was isolated using the CTAB method followed by chloroform extraction and isopropanol precipitation (*Chen and Ronald, 1999*). Four leaves were pooled for one biological replicate and isolated DNA was diluted to a concentration of 5 ng µl$^{-1}$. *H. arabidopsidis* and *A. thaliana* genomic DNA was quantified by qPCR on a qPCR cycler (CFX96, Bio-Rad) using SYBR Green (Invitrogen, Thermo Fischer Scientific) and GoTaq G2 Polymerase (Promega) using species-specific primers (*Supplementary file 3*). Relative DNA content was calculated using the 2$^{-\Delta\Delta Ct}$ method (*Livak and Schmittgen, 2001*).

## *A. thaliana* gene expression analysis

Total RNA was isolated using a CTAB-based method (*Bemm et al., 2016*). Genomic DNA was removed using DNase I (Sigma-Aldrich) and cDNA synthesis was performed with 1 µg total RNA using SuperScriptIII RT or Maxima H$^-$ RT (Thermo Fisher Scientific). Gene expression was measured by qPCR using a qPCR cycler (Quantstudio5, Thermo Fisher Scientific) and Primaquant low ROX qPCR master mix (Steinbrenner Laborsysteme). Differential expression was calculated using the 2$^{-\Delta\Delta Ct}$ method (*Livak and Schmittgen, 2001*).

## Generation of transgene expression vectors

Plasmids for Arabidopsis transformation were constructed using the plant Golden Gate based toolkit (*Binder et al., 2014*). The coding sequences of *AtWNK2* and *AtAED3* were amplified by PCR from Arabidopsis cDNA, and silent mutations were introduced by PCR in the target sequence of *Hpa*sRNA2 and *Hpa*sRNA90, respectively. For overexpression, *AtWNK2r* was ligated into a binary expression vector with a C-terminal GFP tag under the control of the *LjUBQ1* promoter. *AtWNK2r* and *AtAED3r* were also ligated into a binary expression vector with a C-terminal GFP tag under the control of their native promoters (~2 kb upstream of the translation start site). Promoter function was tested by fusion to 2x*GFP-NLS* and fluorescence microscopy of transiently transformed *Nicotiana benthamiana* leaves. STTM sequences were designed as described previously (*Tang et al., 2012*), and flanks with BsaI recognition sites were introduced. STTM sequences were synthesized as single stranded DNA oligonucleotides (Sigma Aldrich). The strands were end phosphorylated by T4 polynucleotide kinase (NEB), annealed, and cloned into an expression vector under the control of the *pro35S*. The final vector with STTMs for *Hpa*sRNA2, *Hpa*sRNA30, and *Hpa*sRNA90 in a row after each other, a rRNA-derived *Hpa*sRNA, or a scrambled sequence was assembled, respectively. The coding sequence of Csy4 was synthesized (MWG Eurofins) with codon optimization for expression in plants. Cloned Csy4 was flanked with new overhangs for integration in the Golden Gate toolkit by PCR. A fusion of the target sequences of *Hpa*sRNA2 and *Hpa*sRNA90, the target sequence of *AtmiRNA164a*, a scrambled target site, and the target sequence of Csy4 were synthesized as single strands (Sigma Aldrich). The strands were end phosphorylated by T4 polynucleotide kinase (NEB) and annealed. Csy4 was flanked with the respective target sequences and ligated into a vector under the control of the *AtWNK2* promoter by BsaI cut ligation. For the reporter, a Csy4 target sequence was inserted between the Kozak sequence and the start codon of the *GUS* gene and ligated into a vector under the control of the *AtEF1α* promoter. The final binary expression vector was assembled by combination of the Csy4 and the *GUS* vectors by BpiI cut ligation. All cloning primers are listed in *Supplementary file 3*.

## Generation of transgenic Arabidopsis plants

Arabidopsis plants of Col-0 (WT), *atwnk2-2,* and *ataed3-1* were transformed with the respective construct using the *Agrobacterium tumefaciens* strain AGL1 by the floral dip method (*Clough and Bent, 1998*). Transformed plants were selected on ½ MS + 1% sucrose agar plates containing 50 µg/ml kanamycin, and were subsequently transferred to soil. Experiments were carried out on T1 generation plants representing independent transformants, unless a transformation line number is indicated (e.g. STTM #4). These experiments were carried out using T2 plants.

## AGO Western blot analysis and sRNA co-immunopurification

SRNAs bound to *A. thaliana* AGO1 or HA-tagged *At*AGO2 were co-immunopurified (co-IPed) from native proteins without any cross-linking agent and isolated as described previously, with minor modifications (*Zhao et al., 2012*). In brief, 5 g infected leaf tissue were ground in liquid $N_2$ to fine powder and thawed in 20 ml IP extraction buffer (20 mM Tris-HCl, 300 mM NaCl, 5 mM $MgCl_2$, 0.5% (v/v) NP40, 5 mM, one tablet/50 ml protease inhibitor (Roche Diagnostics), 200 U RNAse inhibitor (RiboLock, Thermo Fisher Scientific)). The cellular debris was removed by centrifugation at 4000 g and 4°C and the supernatant was filtered with two layers or Miracloth (Merck Millipore). 1 µg α-AGO1 antibody (Agrisera)/g leaf tissue or 0.1 µg α-HA antibody (3F10, Roche or 12CA5)/g leaf tissue was incubated on a wheel at 4°C for 30 min. Protein pull down and washing was performed using 400 µl Protein A agarose beads (Roche) as described by *Zhao et al., 2012*. For Western blot analysis 30% of the co-IP fraction were used, and protein was detected using α-AGO1 antibody (Agrisera) in 1:4000 dilution or α-HA antibody (3F10, Roche or 12CA5) in 1:1000 dilution, respectively. This was followed by an incubation with adequate secondary antibody (α-rabbit IRdye800 (LI-COR, 1:3000 dilution), α-mouse IRdye800 (LI-COR, 1:15000 dilution), and α-rat IRdye800 (LI-COR, 1:15000 dilution)), and protein detection was performed with the Odyssey imaging system (LI-COR). Recovery of the co-IPed sRNAs was achieved as previously described (*Carbonell et al., 2012*), and was directly used for stem-loop RT-PCR analysis or sRNA library preparation.

## Stem-loop RT PCR

SRNAs were detected by stem-loop RT-PCR from 1 µg of total RNA or 5% of the *At*AGO co-IPed RNA, as described previously (*Varkonyi-Gasic et al., 2007*).

## 5' RACE-PCR

5' RACE-PCR was performed on 1 µg of total RNA isolated from *Hyaloperonospora*-infected Arabidopsis leaves pooled from equal amounts isolated at 4 and 7 dpi, using the 5'/3' RACE Kit, 2nd Generation (Roche Diagnostics). After the first round of PCR, a gel fraction of the expected size was cut out and a nested PCR was carried out on the eluted DNA. Bands were cut out and DNA was eluted using GeneJet Gel Extraction Kit (Thermo Fisher Scientific). A library was constructed from the eluted PCR fragments using the sparQ DNA Library Prep Kit (Quantabio) and sequenced on an Illumina MiSeq platform.

## sRNA cloning, sequencing and target gene prediction

SRNAs were isolated from total RNA for high throughput sequencing as previously described (*Weiberg et al., 2013*). SRNAs were cloned for Illumina sequencing using the Next Small RNA Prep kit (NEB) and sequenced on an Illumina HiSeq1500 platform. The Illumina sequencing data were analysed using the GALAXY Biostar server (*Giardine et al., 2005*). Raw data were de-multiplexed (Illumina Demultiplex, Galaxy Version 1.0.0) and adapter sequences were removed (Clip adaptor sequence, Galaxy Version 1.0.0). Sequence raw data are deposited at the NCBI SRA server (BioProject accession: PRJNA395139). Reads were then mapped to a master genome of *Hyaloperonospora arabidopsidis* comprising the isolates Emoy2 (BioProject PRJNA30969), Cala2 (BioProject PRJNA297499), Noks1 (BioProject PRJNA298674) using the BOWTIE algorithm (Galaxy Version 1.1.0) allowing zero mismatches (-v 0). Subsequently, reads were cleaned from *Arabidopsis thaliana* sequences (TAIR10 release) with maximal one mismatch. For normalization, ribosomal RNA (rRNA), transfer RNA (tRNA), small nuclear RNAs (snRNAs), and small nucleolar RNA (snoRNA) reads were filtered out using the SortMeRNA program (Galaxy Version 2.1b.1). The remaining reads were counted and normalized on total *H. arabidopsidis* reads per million (RPM). The *Hpa*sRNAs were clustered if their 5' end position or 3' end position were within the range of three nucleotides referring to the genomic loci (*Weiberg et al., 2013*). Target gene prediction of sRNAs was performed with the TAPIR program using a maximal score of 4.5 and a free energy ratio of 0.7 as thresholds (*Bonnet et al., 2010*). Allelic variation analysis of *Hpa*sRNA target sites in *A. thaliana* mRNAs was done at the 1001Polymorph browser (https://tools.1001genomes.org/polymorph/).

## DNA alignment

Search for homologous sequences of *Hpa*sRNA was performed by BLASTn search using the genomes of Noco2 (PRJNA298674), Cala2 (PRJNA297499) and Emoy2 (PRJNA30969), or the Ensembl Protists database (http://protists.ensembl.org). Homolog DNA sequences of 100 nucleotides up- and downstream of *SRNA2* homologs were aligned using the CLC Main Workbench package.

## Statistical analysis

All statistical tests were carried out using R studio (version 1.0.136, rstudio.com). ANOVA tests were performed on log-transformed data. Letters indicate groups of statistically significant difference by ANOVA followed by TukeyHSD with p≤0.05. The dashes on the letters imply an independent ANOVA with TukeyHSD per time point.

## Acknowledgements

The authors thank Michaela Pagliara for excellent technical assistance, Alexandra Corduneanu for help with data collection of *P. capsici* inoculations, Dr. Martin Parniske for critical reading of the manuscript, inspiring scientific discussions, and support, as well as Christopher Alford for reviewing the manuscript as a native English speaker. We want to thank Dr. Aline Banhara and Fang-Yu Hwu for introducing us into the *H. arabidopsidis*/Arabidopsis pathosystem. We want to thank the Gene Center Munich for Illumina HiSeq sequencing, as well as Gisela Brinkmann and the Genomics Service

Unit of the LMU for Illumina MiSeq service. Seeds used in this study were provided by the Nottingham Arabidopsis Stock Centre (NASC) unless otherwise specified. We thank Dr. Hervé Vaucheret, Dr. James Carrington, and Dr. Steven Jacobsen for kindly providing us seeds of the *atago1-27*, *atdcl2dcl3dcl4*, *atrdr6-15*, *atse-2*, and *proHA:HA-AGO2* mutants and Dr. Tino Köster for the *atdcl1-11* mutant. We thank Dr. Michael Boshart for providing us αHA (12CA5) antibody. We thank Dr. David Chiasson and Martin Bircheneder for providing Golden Gate entry plasmids and Dr. Dagmar Hann for providing the *Pst* DC3000 strain. This work was supported by the German Research Foundation (DFG; Grant-ID WE 5707/1–1). The funders had no role in study design, data collection and analysis, decision to publish or in preparation of the manuscript.

## Additional information

### Funding

| Funder | Grant reference number | Author |
|---|---|---|
| Deutsche Forschungsgemeinschaft | WE 5707/1-1 | Arne Weiberg |

The funders had no role in study design, data collection and interpretation, or the decision to submit the work for publication.

### Author contributions

Florian Dunker, Conceptualization, Data curation, Formal analysis, Validation, Investigation, Methodology, Writing - original draft, Writing - review and editing; Adriana Trutzenberg, Jan S Rothenpieler, Sarah Kuhn, Formal analysis, Investigation; Reinhard Pröls, Methodology; Tom Schreiber, Alain Tissier, Eric Kemen, Ralph Hückelhoven, Resources; Ariane Kemen, Formal analysis, Methodology; Arne Weiberg, Conceptualization, Resources, Data curation, Supervision, Funding acquisition, Validation, Investigation, Methodology, Writing - original draft, Project administration, Writing - review and editing

### Author ORCIDs

Florian Dunker https://orcid.org/0000-0003-1586-412X
Jan S Rothenpieler http://orcid.org/0000-0001-8892-8230
Arne Weiberg https://orcid.org/0000-0003-4300-4864

### Decision letter and Author response

Decision letter https://doi.org/10.7554/eLife.56096.sa1
Author response https://doi.org/10.7554/eLife.56096.sa2

## Additional files

### Supplementary files

- Supplementary file 1. sRNA read numbers.
- Supplementary file 2. Predicted *A. thaliana* target genes of *Hpa*sRNAs.
- Supplementary file 3. List of oligonucleotides used in this study.
- Transparent reporting form

### Data availability

Sequencing data have been deposited in NCBI SRA (PRJNA395139).

The following dataset was generated:

| Author(s) | Year | Dataset title | Dataset URL | Database and Identifier |
|---|---|---|---|---|
| Weiberg A | 2017 | Arabidopsis thaliana Col-0 infected | https://www.ncbi.nlm. | NCBI Sequence Read |

| | | |
|---|---|---|
| with Hyaloperonospora arabidopsidis Noco2 Raw sequence reads | nih.gov/sra/ PRJNA395139 | Archive, PRJNA395139 |

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
