## [Decision Letter]

**Acceptance summary:**

The manuscript deals with the transfer of sRNAs between pathogen and plant hosts and vice versa. The authors report that also oomycetes – which are very different from fungi – exploit small RNA mechanisms to suppress plant defence. Further, a novel CRISPR endoribonuclease Csy4/GUS repressor reporter was developed to visualize in situ pathogen-induced target suppression in *Arabidopsis thaliana*. This first use of an in situ silencing reporter in the context of cross-kingdom RNA interference (ck-RNAi) directly demonstrates the effects of pathogen small RNAs on the host. The author's findings that deletion of the plant dicer-like machinery results in enhanced oomycete burden indicates that the authors discovered a bidirectional ck-RNAi between plants and oomycetes.

**Decision letter after peer review:**

[Editors’ note: the authors submitted for reconsideration following the decision after peer review. What follows is the decision letter after the first round of review.]

Thank you for submitting your work entitled "Oomycete small RNAs invade the plant RNA-induced silencing complex for virulence" for consideration by *eLife*. Your article has been reviewed by three peer reviewers, one of whom is a member of our Board of Reviewing Editors, and the evaluation has been overseen by a Senior Editor. The reviewers have opted to remain anonymous.

Our decision has been reached after consultation between the reviewers. Based on these discussions and the individual reviews below, we regret to inform you that your work in its current form will not be considered further for publication in *eLife*, as we believe that it will be difficult to address all our concerns within two months. Nevertheless, we appreciate the topic and find the work in principle very interesting. Thus, if you choose not to send the work as is elsewhere, but rather revise the study, *eLife* would be prepared to review the work again. It would, however, be treated as a new submission, although we would try to retain the same reviewers.

Summary:

The manuscript reports that oomycetes have similar mechanisms to suppress the plant defence as previously shown for fungal plant pathogens. A novel CRISPR endoribonuclease Csy4/GUS repressor reporter was developed to visualize in situ pathogen-induced target suppression in *Arabidopsis thaliana*. This first use of an in situ silencing reporter in the context of cross-kingdom RNA interference (ck-RNAi) directly demonstrates the effect of pathogen small RNAs on the host. The ck-RNAi phenotype is observed only in plant cells near the oomycete and does not appear to spread beyond these neighboring cells. The author's findings that deletion of the plant dicer-like machinery results in enhanced oomycete burden suggest that this might also be another example of bidirectional ck-RNAi. The authors present a corroborating dataset using RNA-seq *At*AGO1 pulldown.

Essential revisions:

1) In general, some reviewers were concerned about the conceptual novelty of this study.

2) It needs to be indicated whether the *H. arabidopsidis* small RNAs were cross-linked to AGO1 before immunoprecipitation in the Materials and methods. Also, were any controls included using an unrelated protein e.g. with a basic domain that unspecifically binds RNAs. This is highly relevant to show the specificity because the authors identified so many RNAs.

3) Adding data indicating whether other infection with other oomycetes also results in trailing necrosis in the ago1-variants is required to increase the scope of this paper.

4) An STTM construct against sRNA that had no predicted targets is a necessary control.

5) The reporter system described is nice, but confirmation by another method is needed to confirm the value of the results. This could be done by FISH or RNA sequencing.

6) Figure 2: The images are unclear, adding propidium iodide staining of the same mutants is needed to confirm the cell death.

7) Figures 1F-1G: The data for *AtWNK2* are not convincing. Bands are fuzzy and vague, and the RNA ends cloned are nowhere near the scissile phosphate at position 10-11. We do not consider that target verified. If it's due to "alternative sRNAs" (as suggested in the Results), then these alternatives should be shown, and that hypothesis directly tested. As it stands, that target verification is not well supported.

Reviewer #1:

Transfer of sRNAs between pathogen and plant hosts and vice versa has been impressively (and spectacularly) shown for the fungus *Botrytis* and its host plants. The corresponding author of the current *eLife* manuscript was involved in some of this work. The current manuscript is somehow a continuation but also describes novelties. The manuscript reports that oomycetes that are different from fungi have similar mechanisms to suppress the plant defence. Further, a novel CRISPR endoribonuclease Csy4/GUS repressor reporter was developed to visualize in situ pathogen-induced target suppression in *Arabidopsis thaliana*. This first use of an in situ silencing reporter in the context of cross-kingdom RNA interference (ck-RNAi) directly demonstrates the effect of pathogen small RNAs on the host. The ck-RNAi phenotype is observed only in plant cells near the oomycete and does not appear to spread beyond these neighboring cells.

The author's findings that deletion of the plant dicer-like machinery results in enhanced oomycete burden suggest that this might also be another example of bidirectional ck-RNAi, although this was not further pursued. The authors present a corroborating dataset using RNA-Seq *At*AGO1 pulldown.

General comments

The paper is technically sound, but it would benefit from additional editing for grammar and clarification of terms. In particular, for the diverse readership at *eLife*, the manuscript would greatly benefit from better definition of plant specific tools and terms. Conceptionally, the novelty reported is limited. An important paper describing plant defence mRNAs against Oomycestes and their counter strike has not been considered (Hou et al., 2019).

Specific Comments

1) Introduction: The statement that ck-RNAi has only been observed in fungal-plant interactions is not exactly true. Although other studies may not use the term ck-RNAi, I believe that this phenotype has been observed in other instances across kingdoms (with some contention – See PMID 31018602: Zeng J. Cells. 2019. Cross-Kingdom Small RNAs Among Animals, Plants and Microbes).

2) Results section: The sRNA populations of oomycetes have previously been reported to be 21 and 25 nt as the authors also mention, so the language indicating that this is the first example needs to be refined (for example PMID 28512457).

3) It needs to be indicated whether the *H. arabidopsidis* small RNAs were cross-linked to AGO1 before immunoprecipitation in the Materials and methods. Also, were any controls included using an unrelated protein e.g. with a basic domain that unspecifically binds RNAs. This is highly relevant to show the specificity because the authors identified so many RNAs.

4) In the experiments of Figure 1—figure supplement 3, why are the *AtAED3* levels so different between the" WT mock" and "WT infected" at the "before infection" time point? The levels are so much lower for both conditions after infection that it is hard to understand these data. More explanation needs to be added?

5) Would it be possible to pull down the *Hpas*RNAs and detect *At*AGO by western with a tagged STTM construct? The levels might be too low for this to be feasible.

6) In Figure 4—figure supplement 2A, is the "0" symbol indicating wobble pairings? If so, I am not sure if they are all labeled correctly. G-U is generally denoted as a wobble. Also, I would recommend writing U instead of T in the various RNAs shown throughout the text.

7) More detail to plant specific methods and terminology needs to be added to broaden the scope of the manuscript (*R* genes, T-DNA insertion, "debilitated endogenous RNAs", etc). In addition, the authors need to define the GUS reporter at use (or in the Materials and methods).

8) Adding data indicating whether other infection with other oomycetes also results in trailing necrosis in the ago1-variants is required to increase the scope of this paper.

9) A line further describing the STTM concept for Figure 3B would be beneficial.

10) Are the differences in Figure 3—figure supplement 1 significant? I assume not since all other figures include significance?

11) Did the authors ever try to make STTM constructs for a single *Hpas*RNA instead of three? Is it surprising that an STTM construct against only 3 sRNA can have such drastic impact on infection? Do the authors have examples of sRNA that did not shown an effect on infection? Alternatively, an STTM construct against sRNA that had no predicted targets would be a necessary control.

12) The right panel of Figure 4—figure supplement 1A would benefit from slightly more explanation in the legend.

13) The labeling of significance throughout the paper is confusing (What is the difference between a, b, a', b', ab, etc.?). The figure legends also sometimes define the same statistical abbreviation multiple times, which could be condensed for simplicity.

14) “We expected those plant lines to become more resistant against *H. arabidopsidis*” should point out that you expect these strains to be more resistant compared to a T-DNA insertion line, right?

15) How do the findings of Figure 4—figure supplement 4A on the overexpression strains fit with the rest of the model?

16) In Figure 4—figure supplement 4, would measuring oomycete DNA be appropriate for comparisons with the rest of the paper?

17) The manuscript has a lot of supplemental material. It seems like some of the supplemental material should be in the main text (parts of Figure 1—figure supplement 3, Figure 2—figure supplement 1 and Figure 4—figure supplement 1 perhaps?)

18) Do the authors have ideas about how the sRNAs are delivered to the plant? Would it be worth a sentence or two postulating in the discussion?

19) Would it be possible to use FISH to stain for the *A. thaliana* mRNA targets and show that they are decreased in the area around the oomycete? The reporter system described is nice, but confirmation by another method is essential to confirm the results.

20) Can the authors speculate on why *atwnk2-3* did not have a phenotype similar to *atwnk2-2* in Figure 4A?

21) The authors might consider adding a discussion point on the recently published paper PMID 31333714 describing an example of a fungal pathogen that does not undergo ck-RNAi to the end of the Discussion. This new publication might also partially agree with the findings in Figure 2—figure supplement 6? It seems possible that the fungal pathogen tested may also not be using ck-RNAi (see also Hou et al., 2019). Alternatively, it may just utilize a different AGO protein?

22) It is difficult to see the second band in Figure 1F for *AtWNK2*. It would also be more convincing to have the WT and resistant target samples all run together on the same blot to observe the minor shift the authors described in the text.

23) Figure 1—figure supplement 1A would be better in color or with a slightly different color scheme.

24) Can the layout of Figure 1—figure supplement 5 be matched to Figure 1E in in terms of inset orientation and inclusion of numbers of observed phenotypes?

25) “*Hpas*RNA30 was detectable in infected plant leaves at 4 and 7 dpi by stem-loop RT-PCR, but not at 0 and 1 dpi supporting that this sRNA was produced by *H. arabidopsidis* but not by Arabidopsis” – Does this line intend to mean that the sRNA was produced by *H. arabidopsidis* infection but not by uninfected *A. thaliana*?

26) More background info on the selected target genes in the beginning of the Results would allow the reader to appreciate their importance for infection earlier than the end of the discussion.

Reviewer #2:

In the paper "Oomycete small RNAs invade the plant RNA-induced silencing complex for virulence" the authors present evidence for a cross-kingdom sRNA transfer and functionality between an oomycete (*Hyaloperonospora arabidopsidis*) and a plant (*Arabidopsis thaliana*). In general, this is an interesting work. However, I have few concerns and comments that could improve the paper. I have two particular concerns, (i) the paper is tough to read at some points, making difficult to follow authors' reasoning and conclusions, and (ii) the lack of biological replicates for the small RNA libraries could explain the high number of *Hpas*RNAs targeting Ath genes.

Major comments:

– I'm not very supportive of the "invasion" terminology used in the title and in several places in the manuscript, as it implies a mechanism of AGO loading, and this mechanism is not described in the manuscript. It could be passive loading due to concentration, and a passive movement is not an invasion.

– Throughout the manuscript, there are claims of priority ("first sRNA transcriptome", "data providing first evidence that", etc.) that should be removed as most journals don't allow this.

– It would be interesting to know the genomic origin of the *Hpa* 21-mers found in the AGO1-IP samples. Do they predominantly derive from un-annotated regions?

– Also, there's no analysis of the genomic sources of the three small RNAs that are used the most in this work, *Hpas*RNA2, *Hpas*RNA30 and *Hpas*RNA90. What are the source loci?

– For the construct with the miR164 target sites flanking Csy4, why is there not constitutive GUS production? miR164 should be expressed in the leaves.

– Could authors specify the sequence of the alternative *Hpas*RNA90/*Hpas*RNA2 found in their libraries and the abundance for each of the libraries? Were they also found in AGO1-IP libraries? Do the authors hypothesize that the *Hpas*RNAs that are cleaving the Ath targets are different from those that they identified in the AGO1-IP libraries, but come from the same "sRNA precursor"? It is not clear in the text what the hypothesis is.

– Several images in figures contain numbers on the upper right corner. However, it is unclear what these numbers are. Maybe the number of images that look like this out of the total? But then why are several zero? Please explain these in the figure legends and the text.

– Figure 1G: Why is the cleavage not between the 10th and 11th nucleotides of the small RNA, in either case? This is atypical, and perhaps indicative that this small RNA is not the cause of the cleavage. Also, the number of captured cleavage events is very low. This is a case in which degradome or PARE sequencing would be far more convincing.

– Figure 2: I think it would be informative if the authors could label the structures observed in the images. What are *Hpa* haustorium?

– Figure 2: The images are unclear to me. I suggest to adding propidium iodide staining of the same mutants to confirm the cell death.

– Figure 2—figure supplement 4: It would be interesting to have also the *ago1-27* mutant in mock conditions, to be able to make a full comparison.

– Figure 3: For clarity, I suggest adding a representation of the alignment of the *Hpas*RNAs and the STTM sRNAs next to Figure 3A.

– Figure 3—figure supplement 1A: I suggest including all the time points in each graph to show that the de-repression only occurs at the mentioned time point. Also, it would be nice to have line #5 for comparison.

– Subsection “Arabidopsis target genes of *Hyaloperonospora* sRNAs contribute to plant defence”: I do not understand why the mutant lines complemented with the native genes under their own promoter do not behave like WT.

– Discussion: most of the discussion is more a summary of the work presented in the Results section rather than a discussion of the results in the context of the available literature. It is also hard to read at some points (i.e.: "*At*AGO was a major RISC that was hijacked by *Hpas*RNAs to success infection, because both blocking *Hpas*RNAs by transgenic target mimics and dysfunctional *atago1* mutant alleles displayed a clear disease resistance phenotype").

Reviewer #3:

This manuscript analyzes small RNAs from the oomycete plant pathogen, *Hyaloperonospora arabidopsidis*, that seem to attack host plant mRNAs. Overall, a convincing case for pathogen to plant small RNA activity is made, with one interaction demonstrated very clearly, with the possibility of several others. Arabidopsis ago1 hypomorphic mutants support increased pathogen growth, while other Arabidopsis mutants in small RNA biogenesis and function do not; this is consistent with the idea that pathogen sRNAs enter the host and use the host's AGO1 protein to do their damage. Transgenic hosts designed to inhibit 3 of the pathogen small RNAs show increased resistance. Knockout mutants of two potential host mRNA targets show increased susceptibility, also consistent with the overall hypothesis. Complementation of one of these mutants (*aed3*) with a small RNA-resistant allele shows that the presence of the complementary site affects pathogen growth. A clever reporter system, that reports on two pathogen small RNAs at once, also provides convincing evidence of in vivo pathogen sRNA activity against host mRNAs. The work focuses on two host plant targets and corresponding pathogen small RNAs. The data for one of them (*AED3*) is quite convincing, but less so for the other (*WNK2*). Small RNA sequencing suggests there may be many more *H. arabidopsidis* small RNAs that target host mRNAs.

Overall this is an interesting manuscript and a good step forward for the RNAi and plant pathogenesis fields. There are some open questions, some areas that could do with better controls, and a few claims that I feel are unfounded, but these are relatively minor in the context of the overall findings.

Specific Comments:

1) Should there be some kind of control for the AGO1-IP? A no-Ab IP, for instance, or an IP against an irrelevant protein, to assess background levels of contamination? Without this it's quite possible that many of these are non-specific interactions. This doesn't really impact the two small RNAs that were focused on with all of the subsequent focused experiments, but it does seem important to substantiate the claim that there are many of these invading small RNAs that become associated with host AGO1.

2) Figures 1F-1G: The data for *AtWNK2* are not convincing to me. Bands are fuzzy and vague, and the RNA ends cloned are nowhere near the scissile phosphate at position 10-11. I do not consider that target verified. If it's due to "alternative sRNAs" (as suggested in the Results), then these alternatives should be shown, and that hypothesis directly tested. As it stands, that target verification is not well supported.

3) More evidence against *WNK2*: Figure 4D shows no effect of presence or absence of the proposed sRNA target site. And the reporter (Figure 1), and STTM experiments (Figure 3) are designed against multiple pathogen small RNAs, so the contribution, if any, of the single small RNA that might target *WNK2*, is unknowable.

4) Figure 2H/subsection “Arabidopsis *atago1* exhibited enhanced disease resistance against downy mildew”: It seems that there is more pathogen in the rdr mutant and dcl triple mutant. That is consistent with prior work in *Phytophthora* that showed that plant secondary siRNAs attack oomycete genes. I think that interpretation and that prior work should be mentioned here. This is an alternative interpretation to the secondary siRNA –> *R* gene connection that is currently discussed by the authors.

5) “By Arabidopsis *At*AGO1-IP coupled to sRNA-seq, we identified 34 *H. arabidopsidis* sRNAs that hijacked the host RNAi machinery to target multiple plant genes for silencing”: This is not true. Only two small RNAs were directly shown to target plant genes, not 34. All of the others remain merely untested predictions.

6) Figure 4—figure supplement 6B and related text (Discussion paragraph three): I disagree. This is in fact very poor evidence that the *WNK2* complementary sites to this sRNA are conserved – the bottom three alignments are almost certainly non-functional given current understanding of the base-pairing requirements for plant small RNA function.

7) Introduction “Cross-kingdom RNA interference (ck-RNAi) has been reported so far only in fungal-plant interaction” and Discussion “Our study demonstrates the invasion, function, and significance of *Hyaloperonospora* sRNAs in virulence, the first natural ck-RNAi case ever reported for an oomycete plant pathogen”: I'm not sure about this claim. I think Wenbo Ma's work has already shown plant-to-oomycete small RNA activity. If true, rephrase please, because I think "cross kingdom" RNAi from plant to oomycete already in the literature. It would be more accurate to say this is the first report of oomycete –> to plant RNAi transfer, as the reverse (plant –> oomycete) has previously been demonstrated. Indeed, I was very surprised that this prior work in *Phytophthora* was not discussed or cited at all.

[Editors’ note: further revisions were suggested prior to acceptance, as described below.]

Thank you for submitting your article "Oomycete small RNAs bind to the plant RNA-induced silencing complex for virulence" for consideration by *eLife*. Your article has been reviewed by three peer reviewers, one of whom is a member of our Board of Reviewing Editors, and the evaluation has been overseen by Christian Hardtke as the Senior Editor.

The reviewers have discussed the reviews with one another and the Reviewing Editor has drafted this decision to help you prepare a revised submission. The reviewers feel that your manuscript has been considerably improved and describes a very nice study which gives a lot of new insight. Still, there remains some aspects that need your attention (see below).

In recognition of the fact that revisions may take longer than the two months we typically allow, until the research enterprise restarts in full, we will give authors as much time as they need to submit revised manuscripts.

Summary:

The manuscript reports cross-kingdom RNA interference (ck-RNAi) in the interaction of oomycetes that are very different from fungi and plants which leads to suppression of the plant defence. Further, a novel CRISPR endoribonuclease Csy4/GUS repressor reporter was developed to visualize in situ pathogen-induced target suppression in *Arabidopsis thaliana*. This first use of an in situ silencing reporter in the context of ck-RNAi demonstrates the presence of pathogen small RNAs in host cells. The ck-RNAi phenotype was only observed in plant cells near the oomycete and was also specific for the oomycete investigated. The author's findings that deletion of the plant dicer-like machinery results in enhanced oomycete burden suggest that this represents an interesting example of oomycete-to-plant ck-RNAi.

Essential revisions:

1) The one item of some significance remains the 5'-RACE data and more specifically the conclusions being drawn from those data. Figure 1F-H and associated text: The reviewers still feel the data shown do not support the conclusions being made. These data are not strong enough to conclude slicing for either mRNA. We are aware that the previous studies cited (Cai et al., 2018; Zhang et al., 2016) have concluded slicing based on 5'-ends that are not at position 10/11. But just because other studies have made such conclusions does not mean that they were correct. We are aware of no biochemical evidence whatsoever that shows that *At*AGO1 ever can cut anywhere but position 10/11. One explanation in some cases could be that there are "isomiRs" (positional variants of the sRNAs), but the authors backed off of that claim (correctly, since there's no evidence presented). We really don't think these data can be used to draw any firm conclusions and suggest they be struck from the study or presented/discussed as inconclusive rather than conclusive data. This does not substantially affect the overall conclusions of the whole study, which we feel are very convincing based on all the other data shown.

2) Supplementary file 2 holds the predicted targets. How conserved are these targeted sequences among Arabidopsis accessions?

Other critical points:

1) The PR1 and PDF1.2 expression analysis shows no difference. We think the choice of these genes is not optimal as clearly the plant mounts a defense? response in ago1 mutants visible through the trailing necrosis. So, what is activated then?

2) We are considering the *P. capsici* experiments as suboptimal. The images show sporangia and at 48-72hpi most host tissue is likely dead and therefore unable to activate GUS.

3) We are also concerned by the choice of the *WNK2* promoter to drive the Csy4 reporter. The authors state that transcript levels of *WNK2* are altered in compatible vs incompatible interactions. Is it clear that the transcript levels are altered post-transcriptionally and that the promoter itself shows the same responsiveness in different infection scenarios?

4) Have the authors addressed whether the *Hpa* sRNAs could also target *Hpa* transcripts or are they exclusive to plant transcripts? I could not find such data.

5) Point 4 in the previous points from reviewers is still valid and, in our view, using bacteria as an additional system is debatable.

---

## [Author Response]

[Editors’ note: the authors resubmitted a revised version of the paper for consideration. What follows is the authors’ response to the first round of review.]

Essential revisions:1) In general, some reviewers were concerned about the conceptual novelty of this study.

We are convinced that our study provides conceptual novel insights into the mechanism of cross-kingdom RNAi. First, we here provided data that expand the concept of cross-kingdom RNAi as a virulence strategy into a new pathogen class, that of the oomycetes. Cross-kingdom RNAi events have been found in fungal, plant, and animal systems. However, oomycetes have been estimated to have diverged from the plant, animal or fungal clades over 1.5 billion years ago (Parfrey et al., 2011). Under this aspect, we suggest that cross-kingdom RNAi is either evolutionary deeply conserved among distinct eukaryotic pathogens, or it has been independently evolved multiple times. Given that, we feel that our manuscript contains clearly conceptual novelty and it is noteworthy that cross-kingdom RNAi occurs in phylogenetic very distinct organismic interactions.

From the technical point of view, we would also like to point out that we provide a novel type of cross-kingdom silencing reporter in this manuscript. This reporter is based on an inverse readout, meaning that our reporter is switched on by the action of a pathogen small RNA entering a host plant cell and RNAi machinery. This is superior to other cross-kingdom RNAi reporters that typically quantify reduction of target gene expression, for instance by fusion to a fluorescence reporter (e.g. GFP). Using our novel and unique cross-kingdom RNAi reporter, we are able to visualize, which plant cells experience pathogen small RNA-induced gene silencing. With this reporter, we provide conceptual new insights into the spatial-temporal effects of cross-kingdom RNAi in the infected host plant tissue during pathogen attack. We foresee that this novel class of cross-kingdom RNAi reporter will be widely used in future research on inter-kingdom RNA communication.

2) It needs to be indicated whether the H. arabidopsidis small RNAs were cross-linked to AGO1 before immunoprecipitation in the Materials and methods. Also, were any controls included using an unrelated protein e.g. with a basic domain that unspecifically binds RNAs. This is highly relevant to show the specificity because the authors identified so many RNAs.

We thank the reviewer for this important suggestion. We have now implemented a new paragraph describing the applied protocol in greater detail in the Materials and methods section. We now state that there was no cross-linking for *At*AGO-IP coupled to small RNA analysis. Regarding an independent control experiment that rules out any unspecific small RNA binding to Arabidopsis *At*AGO1, we have performed an additional experiment, in which we in parallel pulled down *At*AGO1 and *At*AGO2 for small RNA deep sequencing using *Hyaloperonospora*-infected leaf materials. We decided on *At*AGO2 as our control, because *At*AGO2 is a small RNA binding protein in Arabidopsis, but typically binds types of plant small RNAs different to *At*AGO1 (Mi et al., 2008). AGO2 as a control were also used in fungal-plant cross-kingdom RNAi studies (Weiberg et al., 2013, Wang et al., 2013). Indeed, our new data indicate enriched binding of the *Hyaloperonospora* small RNAs under investigation to *At*AGO1 compared to *At*AGO2. This is in line with the *At*AGO1 and *At*AGO2 pull-down coupled to small RNA RT-PCR results, as presented in Figure 2—figure supplement 2. We now have incorporated the new data of *Hyaloperonospora* small RNA reads found in *At*AGO1 or *At*AGO2 into the Supplementary file 2. The new sequencing data of *At*AGO1 and *At*AGO2 co-IP will be added to the NCBI SRA database, upon acceptance of the manuscript.

3) Adding data indicating whether other infection with other oomycetes also results in trailing necrosis in the ago1-variants is required to increase the scope of this paper.

We thank the reviewer for this suggestion. We have now investigated and compared the infection phenotypes of Arabidopsis wild type and *ago1* mutant plants after inoculation with another oomycete, white rust pathogen *Albugo laibachii*. This infection assay was repeated three times. In none of these could we detect any obvious phenotypic differences between wild type and *atago1-27* plants, and we also did not observe any trailing necrosis in *atago1-27*. Based on this result, we believe that trailing necrosis occurring in the Arabidopsis *ago1* mutants is rather specific to *H. arabidopsidis* infection. We would like to emphasize that this result does not exclude the possibility of cross-kingdom RNAi in the Arabidopsis-*Albugo* interaction. The new data on *Albugo* infection phenotypes are now described in subsection “Arabidopsis *atago1* exhibited enhanced disease resistance against downy mildew” and incorporated into the new Figure 2—figure supplement 6C and D.

4) An STTM construct against sRNA that had no predicted targets is a necessary control.

We understand and share the reviewer’s concern about the possibility that our STTM RNA could have imposed a general immune reaction in plants, and thus this experiment requires a necessary control. In order to minimize the chance of the STTM affecting general immune reaction, we have inoculated our STTM lines with the bacterial pathogen *Pseudomonas syringae* pv *tomato* strain DC3000 as a control, that does not possess a canonical RNAi pathway. In this experiment, we did not observe any increased resistance of STTM plants compared to wild type. That means STTM expressing plants confer resistance exclusively to *H. arabidopsidis*. We now have included description of the bacterial pathogen data in subsection “*Hpas*RNAs are crucial for virulence” and presented them in the new Figure 3D. We decided to not construct additional new STTMs against *Hyaloperonospora* small RNAs for plant transformation, which were not predicted to target plant genes, because this would not exclude the possibility of plant gene silencing, as we chose stringent criteria for target prediction to minimize the number of false positives.

5) The reporter system described is nice, but confirmation by another method is needed to confirm the value of the results. This could be done by FISH or RNA sequencing.

We thank the reviewer(s) for the comments to confirm our novel reporter system. We have undertaken two additional experiments to further validate the strength of our reporter. First, specific activation of *GUS* upon *H. arabidopsidis* infection was confirmed by quantitative RT-PCR. However, we have experienced the same only moderate effects in *GUS* up-regulation similar to down-regulation of Arabidopsis target mRNAs *AtWNK2* and *AtAED3* by *H. arabidopsidis* infection. Thanks to our novel reporter system we now understand that the effect of pathogen small RNAs is restricted to plant cells under direct infection by *H. arabidopsidis*. We have not included the *GUS* expression data into the manuscript, because we think that the data do not provide any additional information to the readership. However, we are happy to share *GUS* expression data as Author response image 1. Upon request, we can include the data in the manuscript. In a second experiment, we have challenged our reporter plants with the hemibiotrophic oomycete pathogen *Phytophthora capsici*. As displayed in Figure 4—figure supplement 6, *Hpa*sRNA2 is conserved among oomycetes, but the additional mismatches in the seed region and the predicted cleavage position make efficient silencing highly unlikely as also pointed out by reviewer 3. We observed GUS activation in few, individual plant cells, but these were independent from infection sites. This result supports the specificity of the reporter system to individual pathogen small RNAs (in our case from *H. arabidopsidis*) that are functional to enter plant cells and induce cross-kingdom RNAi. We now have incorporated phenotype data of *P. capsici* infection into the new Figure 1—figure supplement 5B.

**Author response image 1. sa2fig1:** Relative GUS mRNA levels in cross-kingdom RNAi reporter plants upon water treatment or *H. arabidopsidis* inoculation at 5 dpi. *AtActin* was used as a reference gene. Expression was measured in four independent biological replicates.

We believe that confirming experiments suggested by one reviewer, such as single cell RNA sequencing and RNA FISH, are beyond the scope of this study, as these experiments are highly challenging and cannot be provided by our laboratory in reasonable time. Just to share some thoughts about this, cell-specific RNA-seq is a very challenging experiment, as *H. arabidopsidis* is growing mainly through the leaf intercellular space. Laser micro-dissection was indeed successfully used to collect plant epidermis cells under direct fungal infection by e.g. powdery mildew or rust (Chandran et al., 2010; Hacquard et al., 2010). However, we are not aware of any protocol available or any lab in the world that succeeded in collecting the leaf parenchyma cells that are in direct contact with *H. arabidopsidis* cells, with keeping the RNA intact. Moreover, the most solid visualization of *H. arabidopsidis* infection is Trypan Blue staining involving leaf fixation with lactophenol that is not compatible with any downstream RNA analysis. Given the fact that cross-kingdom RNAi reporter, *At*AGO1-IP, plant *ago1* mutant phenotypes, and STTM plants are independent experiments that all contribute to or provide direct evidence for pathogen small RNA-induced plant gene silencing, let us conclude that cross-kingdom RNAi is a virulence strategy of *H. arabidopsidis*.

6) Figure 2: The images are unclear, adding propidium iodide staining of the same mutants is needed to confirm the cell death.

We appreciate this suggestion. We have now used Propidium Iodide staining of wild type and *ago1* mutant plants at 7 dpi with *H. arabidopsidis*. In summary, microscopic examination revealed fluorescence activity in *ago1* mutants, probably indicating dead plant cells, however to us the pictures are less informative than Trypan Blue staining, because it was impossible to localize *H. arabidopsidis* in the Propidium Iodide assay. To our knowledge, Trypan Blue staining is a well-established method to visualize the pathogen structure and it can detect local plant cell death upon *H. arabidopsidis* infection in parallel (Knoth and Eulgem, 2008; van Wees, 2008). Representative Propidium Iodide microscopic images are shown in Author response image 2, but have not been included in the manuscript.

**Author response image 2. sa2fig2:** Arabidopsis wild type, *atago1-27* and *atago1-45* plants were stained with Propidium Iodide at 7 dpi with *H. arabidopsidis*. Fluorescence signals were detected with a rhodamine filter set and are shown in yellow. A minimum of 5 leaves was inspected per genotype with comparable results.

7) Figures 1F-1G: The data for AtWNK2 are not convincing. Bands are fuzzy and vague, and the RNA ends cloned are nowhere near the scissile phosphate at position 10-11. We do not consider that target verified. If it's due to "alternative sRNAs" (as suggested in the Results), then these alternatives should be shown, and that hypothesis directly tested. As it stands, that target verification is not well supported.

We agree with the reviewer that RACE-PCR bands are not perfect in showing a clear cleavage product for *AtWNK2*. However, sequencing PCR products revealed that *Hpa*-infected wild type plants indeed contained mRNA 5` ends that fall into the predicted cleavage site. To make these more obvious, we have repeated the whole experiment by isolating fresh RNA from infected plants and performed another independent RACE-PCR and sequence analysis for *AtWNK2*. By this, we could identify additional clones confirming cleavage at the predicted target site of *Hpa*sRNA2. The numbers have now been incorporated into updated Figure 1H. As pointed out by the reviewer, our sequencing results indicated cleavage of *AtWNK2* at position 11/12, 12/13, which is not the exact position of 10/11. We have stated this observation in the manuscript text. We removed our comment that this slight shift might be indicative for alternative *Hpa*sRNAs, which we agree with the reviewer, is not supported by the accompanying experiments in this manuscript.

However, we would like to emphasise that unexpected shift of the cleavage position had been also reported in previous plant-fungal cross kingdom RNAi studies (Cai et al., 2018; Zhang et al., 2016).

Reviewer #1:[…]Specific Comments1) Introduction: The statement that ck-RNAi has only been observed in fungal-plant interactions is not exactly true. Although other studies may not use the term ck-RNAi, I believe that this phenotype has been observed in other instances across kingdoms (with some contention – See PMID 31018602: Zeng J. Cells. 2019. Cross-Kingdom Small RNAs Among Animals, Plants and Microbes).

We have corrected this sentence and acknowledge correctly recently published work to this subject. The text now reads: “The exchange of small RNAs between host and pathogen can lead to functional gene silencing in the recipient organism, a mechanism termed cross-kingdom RNAi (ck-RNAi). While fungal small RNAs promoting virulence is relatively well established, it is not clear how conserved and significant ck-RNAi is for virulence of distinct plant pathogens.”

2) Results section: The sRNA populations of oomycetes have previously been reported to be 21 and 25 nt as the authors also mention, so the language indicating that this is the first example needs to be refined (for example PMID 28512457).

We have removed the sentence, claiming a “first report” to avoid confusion. We now have included the suggested reference next to the Fahlgren et al. citation.

3) It needs to be indicated whether the H. arabidopsidis small RNAs were cross-linked to AGO1 before immunoprecipitation in the Materials and methods. Also, were any controls included using an unrelated protein e.g. with a basic domain that unspecifically binds RNAs. This is highly relevant to show the specificity because the authors identified so many RNAs.

We have addressed this comment in the essential revision under point 2. In brief, we now describe that *At*AGO pull-down experiments were done without any cross-linking in the Materials and methods section. We performed a new experiment of *At*AGO1 and *At*AGO2-IP sRNA-seq and we now provide new data (Supplementary file 2) that validate *Hyaloperonospora* small RNAs under investigation were enriched in binding to *At*AGO1.

4) In the experiments of Figure 1—figure supplement 3, why are the AtAED3 levels so different between the" WT mock" and "WT infected" at the "before infection" time point? The levels are so much lower for both conditions after infection that it is hard to understand these data. More explanation needs to be added?

We have now included an additional explanation why expression levels are changed also under mock condition. One explanation might be the high humidity (almost 100%) during the course of infection. We have added a sentence “that might have been caused by the almost 100% relative humidity during the infection assay”.

5) Would it be possible to pull down the HpasRNAs and detect AtAGO1 by western with a tagged STTM construct? The levels might be too low for this to be feasible.

We thank the reviewer for this intriguing idea. However, we agree with the reviewer that this is not feasible.

6) In Figure 4—figure supplement 2A, is the "0" symbol indicating wobble pairings? If so, I am not sure if they are all labeled correctly. G-U is generally denoted as a wobble. Also, I would recommend writing U instead of T in the various RNAs shown throughout the text.

We wish to thank the reviewer for pointing out the mistakes in the alignment. We now have fixed all of them.

7) More detail to plant specific methods and terminology needs to be added to broaden the scope of the manuscript (R genes, T-DNA insertion, "debilitated endogenous RNAs", etc). In addition, the authors need to define the GUS reporter at use (or in the Materials and methods).

We have made significant efforts to make our manuscript easier to read and understandable to the broad readership of *eLife*. As suggested by this reviewer, we have defined *R* genes and T-DNA insertion lines as : “While *atwnk2-2* and *ataed3-1* are two SALK/SAIL lines (Alonso et al., 2003) that carry a T-DNA insertion in their CDS, respectively, we now re-located the T-DNA insertion of the *atwnk2-3* plant line from the last exon into the 3´ UTR, based on sequencing the T-DNA flanking sites” We also have changed the term “debilitated endogenous RNAs” to “impaired function of endogenous sRNAs”.

8) Adding data indicating whether other infection with other oomycetes also results in trailing necrosis in the ago1-variants is required to increase the scope of this paper.

We have addressed this comment in the essential revision point 3. In brief, we now have investigated the infection phenotype of Arabidopsis *ago1* mutants after inoculation with the white rust pathogen *Albugo laibachii*. We could not detect any trailing necrosis; thus, it seems that this particular phenotype is specific to *H. arabidopsidis* infection. The data are now described in subsection “Arabidopsis *atago1* exhibited enhanced disease resistance against downy mildew” and in the new Figure 2—figure supplement 6C and D.

9) A line further describing the STTM concept for Figure 3B would be beneficial.

We now have added a sentence that describes the STTM lines, as follows: “The triple STTM RNA was designed to bind the target pathogen small RNAs *Hpa*sRNA2, *Hpa*sRNA30, and *Hpa*sRNA90 by base-pairing. An intended 3-base loop structure at the position 10/11 from the 5` ends of the small RNAs was incorporated to block any potential cleavage by plant AGO. The STTM strategy has been used in plants to scavenge small RNAs to prevent gene silencing of their native target genes (Tang et al., 2012).”

10) Are the differences in Figure 3—figure supplement 1 significant? I assume not since all other figures include significance?

We apologize for being unclear. These data do not indicate statistical significance. We now have stated in the figure legend: “The differences of the average are not statistically significant as determined by Student’s t-test.”

11) Did the authors ever try to make STTM constructs for a single HpasRNA instead of three? Is it surprising that an STTM construct against only 3 sRNA can have such drastic impact on infection? Do the authors have examples of sRNA that did not shown an effect on infection? Alternatively, an STTM construct against sRNA that had no predicted targets would be a necessary control.

We are very excited about the observed reduced infection levels in our triple STTM plants. Yet, we have not tested the contribution of single sRNA STTMs. Nevertheless, we have now performed infection assays with the bacterial pathogen *P. syringae* in order to test, if STTMs induce a more general immune response. We can now exclude this possibility, because we did not observe any enhanced resistance of STTM plants against *P. syringae*. These data are now incorporated into the manuscript in Figure 3D.

12) The right panel of Figure 4—figure supplement 1A would benefit from slightly more explanation in the legend.

We have added an explanatory sentence into the legend of Figure 4—figure supplement 1, stating: “Gene models of *AtWNK2* and *AtAED3*. The insertion site of the T-DNA is marked by the triangles and the genotyping primer binding sites are shown with arrows.”

13) The labeling of significance throughout the paper is confusing (What is the difference between a, b, a', b', ab, etc.?). The figure legends also sometimes define the same statistical abbreviation multiple times, which could be condensed for simplicity.

We apologize for the confusion. We thought to use dashes, if ANOVA tests were not to compare between all the treatments, but only between the treatments of a given time point. We have clarified this now in the Materials and methods section. We now also indicate the used statistics at the end of each figure legend in order to avoid repetition. We hope that this brings clarification and simplicity.

14) “We expected those plant lines to become more resistant against H. arabidopsidis” should point out that you expect these strains to be more resistant compared to a T-DNA insertion line, right?

The assumption made by this reviewer is correct. To avoid misunderstandings, we have now clarified this point, which now reads: “We expected those plant lines to become more resistant against *H. arabidopsidis*, compared to the knockout mutant lines *ataed3-1* and *atwnk2-2*.”

15) How do the findings of Figure 4—figure supplement 4A on the overexpression strains fit with the rest of the model?

We interpret the data as further indicator that *AtWNK2* is part in the plant immune system, because overexpression of immunity genes often leads to autoimmune response. We now state in the manuscript text: “*AtWNK2*r-OE plants showed ectopic cell death in distance from infection sites (Figure 4—figure supplement 4A), as previously described for overexpression lines of other immunity factors, such as *At*BAK1 (Domínguez-Ferreras et al., 2015).”

16) In Figure 4—figure supplement 4, would measuring oomycete DNA be appropriate for comparisons with the rest of the paper?

We agree with the reviewer. Nevertheless, given that such data are of minor relevance, we decided to work on the major critics that have been raised by the three reviewers and the editor for a revised version of this manuscript.

17) The manuscript has a lot of supplemental material. It seems like some of the supplemental material should be in the main text (Parts of Figure 1—figure supplement 3, Figure 2—figure supplement 1 and Figure 4—figure supplement 4 perhaps?)

We have carefully chosen most representative and informative data as main figures of the manuscript, and we feel to not overload those with additional data. However, we are totally open for suggestions, which data could be moved from the supplementary section into the main manuscript.

18) Do the authors have ideas about how the sRNAs are delivered to the plant? Would it be worth a sentence or two postulating in the discussion?

We are happy to share our thoughts on the matter of small RNA delivery in the Discussion section. In fact, extracellular vesicles have garnered a lot of attention in the context of small RNA transport during plant-microbe interactions. We therefore discuss about EVs as potential vehicle of small RNAs based on published reports, and now include a statement in the Discussion: “By which pathways and mechanisms *H. arabidopsidis* small RNAs are translocated into plant cells remains an open question. […] In this regard, transfer of plant small RNA into pathogen cells via exosomal vesicles was reported to induce cross-kingdom RNAi (Cai et al., 2018; Hou et al., 2019), making extracellular vesicles a prime suspect for *H. arabidopsidis* small RNA transport into plant cells.”

19) Would it be possible to use FISH to stain for the *A. thaliana* mRNA targets and show that they are decreased in the area around the oomycete? The reporter system described is nice, but confirmation by another method is essential to confirm the results.

We have addressed this comment in the essential revision under point 5.

20) Can the authors speculate on why atwnk2-3 did not have a phenotype similar to atwnk2-2 in Figure 4A?

We believe that the difference might be due to the fact that T-DNA is inserted in the 3’ UTR in *atwnk2-3* plants, and thus might represent a weaker knockout allele. We now have stated this in the text, accordingly.

21) The authors might consider adding a discussion point on the recently published paper PMID 31333714 describing an example of a fungal pathogen that does not undergo ck-RNAi to the end of the Discussion. This new publication might also partially agree with the findings in Figure 2—figure supplement 6? It seems possible that the fungal pathogen tested may also not be using ck-RNAi (see also Hou et al., 2019). Alternatively, it may just utilize a different AGO protein?

We agree with the reviewer to include such a possibility in our discussion part by referring to recent publications. However, we do not feel confident to speculate if plant *ago1* mutants do not exhibit enhanced disease resistance, as in the cases of *Erysiphe* and *Albugo*, those pathogens are not capable to use small RNAs for infection. As already pointed out by the reviewer, this observation could be explained by various reasons, including a balancing effect between the use for pathogen small RNAs and the use of endogenous plant miRNAs and siRNAs to regulate immune activation. Nevertheless, we now discuss on potential absence of cross-kingdom RNAi in individual pathogen systems in the last part of the discussion.

22) It is difficult to see the second band in Figure 1F for AtWNK2. It would also be more convincing to have the WT and resistant target samples all run together on the same blot to observe the minor shift the authors described in the text.

We now provide the original gel pictures in the modified Figure 1F and 1G including a size marker that allow to directly compare band sizes between the wild type plants and plants expressing a target site resistant version in the respective target genes knockout background.

23) Figure 1—figure supplement 1A would be better in color or with a slightly different color scheme.

We now make the color code in Figure 1—figure supplement 1A compatible to Figure 1—figure supplement 1B.

24) Can the layout of Figure 1—figure supplement 5 be matched to Figure 1E in in terms of inset orientation and inclusion of numbers of observed phenotypes?

We now have made the suggested changes in the Figure 1—figure supplement 5, accordingly.

25) “HpasRNA30 was detectable in infected plant leaves at 4 and 7 dpi by stem-loop RT-PCR, but not at 0 and 1 dpi supporting that this sRNA was produced by *H. arabidopsidis* but not by Arabidopsis” – Does this line intend to mean that the sRNA was produced by *H. arabidopsidis* infection but not by uninfected *A. thaliana*?

The assumption of this reviewer is correct. We have clarified this point, as follows: “The *Hpa*sRNA30 sequence mapped only to the *Hyaloperonospora* genome but not the Arabidopsis reference (see Materials and methods), and it was detected only in infected plants at 4 and 7 dpi by stem-loop RT-PCR (Figure 1—figure supplement 2). Thus, we concluded *Hpa*sRNA30 was produced by *H. arabidopsidis* but not by Arabidopsis.”.

26) More background info on the selected target genes in the beginning of the Results would allow the reader to appreciate their importance for infection earlier than the end of the discussion.

We now have added a sentence at the beginning of the result part: “In addition, members of the WNK protein family and extracellular aspartyl proteases have been previously linked to stress responses and immunity (Balakireva and

Zamyatnin, 2018; Cao-Pham et al., 2018).”

Reviewer #2:[…]Major comments:– I'm not very supportive of the "invasion" terminology used in the title and in several places in the manuscript, as it implies a mechanism of AGO loading, and this mechanism is not described in the manuscript. It could be passive loading due to concentration, and a passive movement is not an invasion.

According to the reviewer`s concern, we now have modified the term to avoid any misunderstandings. The title was changed to “Oomycete small RNAs bind to the plant RNA-induced silencing complex for virulence”. Throughout the manuscript text, we exchanged “invade” with either “enter” or “translocate”.

– Throughout the manuscript, there are claims of priority ("first sRNA transcriptome", "data providing first evidence that", etc.) that should be removed as most journals don't allow this.

We thank the reviewer for this hint. We have now changed the respective terminology and removed the claims of priority.

– It would be interesting to know the genomic origin of the Hpa 21-mers found in the AGO1-IP samples. Do they predominantly derive from un-annotated regions?

To provide this valuable information, we now have included a respective column in Supplementary file 2. The data represent all small RNAs mapped to non-annotated regions, with one exception that mapped to an intronic region of a protein coding gene. We have included in the text: “Most of the *At*AGO1-bound *Hpa*sRNAs with predicted Arabidopsis target genes mapped to non-annotated, intergenic regions (Supplementary file 2).”

– Also, there's no analysis of the genomic sources of the three small RNAs that are used the most in this work, HpasRNA2, HpasRNA30 and HpasRNA90. What are the source loci?

These three small RNAs originate from non-annotated, intergenic genomic regions. We now include this information in the Supplementary file 2.

– For the construct with the miR164 target sites flanking Csy4, why is there not constitutive GUS production? miR164 should be expressed in the leaves.

We appreciate this comment. We now provide additional information in our manuscript. In brief, Arabidopsis miR164 expression is refined to the apical meristem in younger, developing leaves at the serrating leaf margins (Nikovics et al., 2006). Our miR164-specific GUS reporter confirms such an expression pattern around the leaf teeth (see Author response image 3). In addition, we also detected GUS activation in some leaf meristem cells. However, these are unrelated to *H. arabidopsis* infection, as we cannot find any pathogen hyphae at these spots in the microscopy images. In addition, the miR164-specific GUS reporter was constitutively activated in adult rosette leaves. We have obtained such results through our infection assay with *P. capsici* (see Author response image 3). We are happy to include the results into our manuscript, upon request.

**Author response image 3. sa2fig3:** *AtmiR164* reporter plants revealed GUS activation at leaf teeth and in mature leaves. a) Plants expressing AtmiR164 target site (ts) reporter displayed GUS activation at the serrated leaf tips and at the leaf tooth. In rare cases (two over all three round of experimental replication), also patchy GUS activity was observed, which was unlinked to pathogen presence. b) In mature plants, GUS activity was visible throughout the leaf in AtmiR164ts reporter plants, here shown with infecting *P. capsici*. However, GUS activity was independent of the pathogen presence. Scale bars represent 50 μm.

– Could authors specify the sequence of the alternative HpasRNA90/HpasRNA2 found in their libraries and the abundance for each of the libraries? Were they also found in AGO1-IP libraries? Do the authors hypothesize that the HpasRNAs that are cleaving the Ath targets are different from those that they identified in the AGO1-IP libraries, but come from the same "sRNA precursor"? It is not clear in the text what the hypothesis is.

We thank the reviewer for pointing out this unclarity. For target gene prediction, the sRNAs were grouped, if their 5’ end position and 3’ end position were within a 3-nucleotides sliding window, but mapped to the same genomic locus, according to Weiberg et al., 2013. Reads with slight differences at 5` and 3` ends were identified throughout all libraries. Thus, prediction of the slicing position was eventually deviating from the expected position 10/11. We now have clarified small RNA definition for target gene prediction within the Materials and methods part. Moreover, we have now removed our comment that this slight shift might be indicative for alternative HpasRNAs, which we agree with the reviewer #3 is not supported by the accompanying experiments in this manuscript.

– Several images in figures contain numbers on the upper right corner. However, it is unclear what these numbers are. Maybe the number of images that look like this out of the total? But then why are several zero? Please explain these in the figure legends and the text.

We apologize for this confusion and we have now clarified this aspect in our figures and manuscript. As already assumed by the reviewer, the digits incorporated into the figures refer to the case number of a certain event, e.g. trailing necrosis or GUS activity, in relation to the total number of the inspected leaves. In case of a zero value, it means that not a single leaf displayed the event. For example, (0/34) in Figure 2A means that not a single leaf out of 34 inspected leaves showed trailing necrosis. We have now specified an explanation at the end of each figure legend that comes together with the applied statistical test in order to improve the understanding of each figure. However, we believe that additional explanation on this subject in the main manuscript would introduce lots of redundancy, in that case we decided to not make changes in the manuscript text.

– Figure 1G: Why is the cleavage not between the 10th and 11th nucleotides of the small RNA, in either case? This is atypical, and perhaps indicative that this small RNA is not the cause of the cleavage. Also, the number of captured cleavage events is very low. This is a case in which degradome or PARE sequencing would be far more convincing.

We noticed this unexpected shift in the expected cleavage sites. One explanation that is already mentioned in the manuscript text, is alternative *H. arabidopsidis* small RNA sequences that we detected in our small RNA-seq data could be responsible for alternative mRNA cleavage position. Slight shifts away from the expected cleavage position were also reported in other studies describing cross kingdom RNAi (Cai et al., 2018; Zhang et al., 2016). Thus, this seems to be a common observation that could be also partially explained by rapid degradation of 5` uncapped mRNA cleavage products. We believe that this circumstance makes it so hard to detect the expected, precise *AtWNK2* mRNA cleavage product in our study. In any case, we have run an independent RACE-PCR experiments extracting fresh RNA from *H. arabidopsis*-infected wild type plants. From this, we sequenced additional clones that confirmed cleavage sites in the predicted target sequence of *HpasiR2*. The data are now incorporated into the updated Figure 1h. In theory, PARE sequencing could give additional information on global mRNA cleavage profiles, however, in this study, our goal was to confirm predicted target mRNA slicing of two Arabidopsis genes.

– Figure 2: I think it would be informative if the authors could label the structures observed in the images. What are Hpa haustorium?

We now have included arrow symbols to point *H. arabidopsidis* haustoria in microscopic images in Figure 2. A respective introduction of the arrow symbols is included in the figure legend, accordingly.

– Figure 2: The images are unclear to me. I suggest to adding propidium iodide staining of the same mutants to confirm the cell death.

We have addressed this comment in the essential revision under point 6. In brief, we now have performed Propidium Iodide staining experiments upon *H. arabidopsidis* inoculation. Results can be found in the Author response image 2.

– Figure 2—figure supplement 4: It would be interesting to have also the ago1-27 mutant in mock conditions, to be able to make a full comparison.

Indeed, it would be interesting to include an *atago1-27* mock condition in this experiment. In our opinion, this condition contributes to a lesser extent to the major scope of this study, and therefore was not included in this revision.

– Figure 3: For clarity, I suggest adding a representation of the alignment of the HpasRNAs and the STTM sRNAs next to Figure 3A.

We have now added the respective alignments as part of a modified Figure 3A.

– Figure 3—figure supplement 1A: I suggest including all the time points in each graph to show that the de-repression only occurs at the mentioned time point. Also, it would be nice to have line #5 for comparison.

We here have focused on respective time points to investigate the effect of STTMs. Since the two data sets come from two independent experiments with the respective time points taken, it is unfortunately not possible to display the entire time course. It would mix up data of two different experimental sets in one graph, which we believe is misleading. We agree that data for STTM #5 would be nice for comparison, but we had to make priorities for the experiments to be feasible in the given time frame and therefore selected only one line for these measurements.

– Subsection “Arabidopsis target genes of Hyaloperonospora sRNAs contribute to plant defence”: I do not understand why the mutant lines complemented with the native genes under their own promoter do not behave like WT.

We apologize for this misunderstanding. In this experiment, we did not use wild type plants as our control, but the respective T-DNA insertion knockout line that we transformed with an empty vector. We believe that this is a more appropriate control than wild type plants, as these also represent transformed individuals. To clarify this point, we now state in: “We expected those plant lines to become more resistant against *H. arabidopsidis*, compared to the knockout mutant lines *ataed3-1* and *atwnk2-2*.”

– Discussion: most of the discussion is more a summary of the work presented in the Results section rather than a discussion of the results in the context of the available literature. It is also hard to read at some points (i.e.: "AtAGO1 was a major RISC that was hijacked by HpasRNAs to success infection, because both blocking HpasRNAs by transgenic target mimics and dysfunctional atago1 mutant alleles displayed a clear disease resistance phenotype").

We thank the reviewer for this comment. We have revised the Discussion section, accordingly, and now cite additional literature to set our findings into the context of current knowledge in the field. Thereby, we hope to provide the readership a valuable discussion and conclusion of our findings.

Reviewer #3:[…]Specific Comments:1) Should there be some kind of control for the AGO1-IP? A no-Ab IP, for instance, or an IP against an irrelevant protein, to assess background levels of contamination? Without this it's quite possible that many of these are non-specific interactions. This doesn't really impact the two small RNAs that were focused on with all of the subsequent focused experiments, but it does seem important to substantiate the claim that there are many of these invading small RNAs that become associated with host AGO1.

We have addressed this comment in the essential revision under point 2. In brief, we now have implemented a new dataset of *At*AGO1 and *At*AGO2-IP RNA-seq at 4 dpi with *H. arabidopsidis*, with *At*AGO2-IP as a control sample. Our data indicates that most *H. arabidopsidis* small RNAs under investigation are enriched in binding to *At*AGO1. We think *At*AGO2 is a valuable control, as it also binds to small RNAs, but to different small RNA populations compared to *At*AGO1 (Mi et al., 2008). Our new *At*AGO sRNA-seq data are now available in Supplementary file 2.

2) Figures 1F-1G: The data for AtWNK2 are not convincing to me. Bands are fuzzy and vague, and the RNA ends cloned are nowhere near the scissile phosphate at position 10-11. I do not consider that target verified. If it's due to "alternative sRNAs" (as suggested in the Results), then these alternatives should be shown, and that hypothesis directly tested. As it stands, that target verification is not well supported.

We have addressed this comment in the essential revision under point 7. We agree with the reviewer that RACE-PCR bands are not perfect in showing a clear cleavage product for *AtWNK2*. However, sequencing PCR products revealed that *Hpa*-infected wild type plants indeed contained mRNA 5` ends that fall into the predicted cleavage site. To make these more obvious, we have repeated the whole experiment by isolating fresh RNA from infected plants and performed another independent RACE-PCR and sequence analysis for *AtWNK2*. By this, we could identify additional clones confirming cleavage at the predicted target site of *Hpa*sRNA2. The numbers have now been incorporated into updated Figure 1H. As pointed out by the reviewer, our sequencing results indicated cleavage of *AtWNK2* at position 11/12, 12/13, which is not the exact position of 10/11. We have stated this observation in the manuscript text. We removed our comment that this slight shift might be indicative for alternative *Hpa*sRNAs, which we agree with the reviewer is not supported by the accompanying experiments in this manuscript. However, we would like to emphasise that unexpected shift of the cleavage position had been also reported in previous plant-fungal cross kingdom RNAi studies (Cai et al., 2018; Zhang et al., 2016).

3) More evidence against WNK2: Figure 4D shows no effect of presence or absence of the proposed sRNA target site. And the reporter (Figure 1), and STTM experiments (Figure 3) are designed against multiple pathogen small RNAs, so the contribution, if any, of the single small RNA that might target WNK2, is unknowable.

We acknowledge the concern of this reviewer. We have now weakened our statement about *AtWNK2* being a target of *Hpa*sRNA2 by RACE-PCR, accordingly. Nevertheless, STTM plants exhibited enhanced mRNA levels of *AtWNK2* compared to EV control plants, strongly suggesting that *Hpa*sRNA2 is able to suppress *AtWNK2*. Moreover, *atwnk2* knockout plants were more susceptible to *Hyaloperonospora* (Figure 4B-C), which was compensated by expressing *AtWNK2* in the mutant background.

4) Figure 2H/subsection “Arabidopsis atago1 exhibited enhanced disease resistance against downy mildew”: It seems that there is more pathogen in the rdr mutant and dcl triple mutant. That is consistent with prior work in Phytophthora that showed that plant secondary siRNAs attack oomycete genes. I think that interpretation and that prior work should be mentioned here. This is an alternative interpretation to the secondary siRNA –> R gene connection that is currently discussed by the authors.

We now have extended the respective paragraph to: “Both mutants did not exhibit either trailing necrosis (Figure 2G) or reduced, but even increased, pathogen biomass (Figure 2H) upon inoculation with *H. arabidopsidis*. This higher susceptibility was also in line with the previously described role of secondary siRNAs in anti-oomycete defence silencing pathogen genes (Hou et al., 2019).” In the light of recently published literature, we have added also a new paragraph in the Discussion about the possibility of this as the first indication for bi-directional cross kingdom RNAi in the Arabidopsis/*H. arabidopsidis* interaction.

5) “By Arabidopsis AtAGO1-IP coupled to sRNA-seq, we identified 34 H. arabidopsidis sRNAs that hijacked the host RNAi machinery to target multiple plant genes for silencing”: This is not true. Only two small RNAs were directly shown to target plant genes, not 34. All of the others remain merely untested predictions.

We now have reworded this statement, accordingly, and it now reads: “Sequencing small RNAs isolated from Arabidopsis AGO1 revealed at least 34 *H. arabidopsidis* sRNAs that entered the host RNAi machinery and potentially target multiple plant genes for silencing.”

6) Figure 4—figure supplement 6B and related text (Discussion paragraph three): I disagree. This is in fact very poor evidence that the WNK2 complementary sites to this sRNA are conserved – the bottom three alignments are almost certainly non-functional given current understanding of the base-pairing requirements for plant small RNA function.

We now have revised our statement, accordingly. We hope the reviewer can agree that the *siR2* sequence is rather conserved in oomycete pathogens, as depicted in Figure 4—figure supplement 6A. The *WNK2* target region and sites of *siR2* lies towards the Nterminal part of the gene that is also more conserved than the C-terminus (see Author response image 4). However, whether individual base pairing between oomycete *siR2* and target sites can still result in gene silencing is speculative, and as rightfully pointed out by reviewer #3 unlikely for the last three alignments shown in the Figure 4—figure supplement 6B. We have therefore rephrased the entire paragraph: “Moreover, the target site of the pathogen *siR2* homologs lies within a conserved region of other plant *WNK2* orthologs, with the lowest number of base pair mismatches occurring in the highly-adapted *A. thaliana*/*H. arabidopsidis* interaction (Figure 4—figure supplement 6B). Whether this is a sign of pathogen adaptation and whether *siR2* plays any role in other oomycete-plant interactions, remains to be further investigated.”

**Author response image 4. sa2fig4:** Alignment of the *WNK2* orthologs from four distinct plant species, Solanum lycopersicum, Nicotiana benthamiana, Glycine max and *Arabidopsis thaliana*. a) N-terminal section, b) full-length gene alignment indicating different levels of nucleotide sequence conservation with black (low) to red (high) sequence identity. The red arrow indicates the *siR2* target site.

7) Introduction “Cross-kingdom RNA interference (ck-RNAi) has been reported so far only in fungal-plant interaction” and Discussion “Our study demonstrates the invasion, function, and significance of Hyaloperonospora sRNAs in virulence, the first natural ck-RNAi case ever reported for an oomycete plant pathogen”: I'm not sure about this claim. I think Wenbo Ma's work has already shown plant-to-oomycete small RNA activity. If true, rephrase please, because I think "cross kingdom" RNAi from plant to oomycete already in the literature. It would be more accurate to say this is the first report of oomycete –> to plant RNAi transfer, as the reverse (plant –> oomycete) has previously been demonstrated. Indeed, I was very surprised that this prior work in Phytophthora was not discussed or cited at all.

We agree that the work by Hou et al., 2019 should be cited, and we now have cited this paper in our Results and Discussion parts, stating that plant defense against oomycetes with small RNAs has been suggested (but not proven) previously. Additionally, we now have stated without a claim of priority: “Our study demonstrates that ck-RNAi occurs during *Hyaloperonospora* host infection, which contributes to the virulence of this pathogen.” Moreover, we have now revised the first two sentences of our Abstract, accordingly, to avoid confusion as suggested by this and the other reviewers. It reads now: The exchange of small RNAs between host and pathogen can lead to functional gene silencing in the recipient organism, a mechanism termed cross-kingdom RNAi (ck-RNAi). While fungal small RNAs promoting virulence is relatively well established, it is not clear how conserved and significant ck-RNAi is for virulence of distinct plant pathogens.

[Editors’ note: what follows is the authors’ response to the second round of review.]

Essential revisions:1) The one item of some significance remains the 5'-RACE data and more specifically the conclusions being drawn from those data. Figure 1F-H and associated text: The reviewers still feel the data shown do not support the conclusions being made. These data are not strong enough to conclude slicing for either mRNA. We are aware that the previous studies cited (Cai et al, 2018; Zhang et al., 2016) have concluded slicing based on 5'-ends that are not at position 10/11. But just because other studies have made such conclusions does not mean that they were correct. We are aware of no biochemical evidence whatsoever that shows that AtAGO1 ever can cut anywhere but position 10/11. One explanation in some cases could be that there are "isomiRs" (positional variants of the sRNAs), but the authors backed off of that claim (correctly, since there's no evidence presented). We really don't think these data can be used to draw any firm conclusions and suggest they be struck from the study or presented/discussed as inconclusive rather than conclusive data. This does not substantially affect the overall conclusions of the whole study, which we feel are very convincing based on all the other data shown.

We understand the remaining significance regarding the 5’RACE-PCR results. We agree that sequencing a few single clones to clarify the cleavage site is not sufficient, and we want to thank for the insightful comments regarding shift of cleavage sites. We now have performed a new 5’RACE-PCR experiment using *Hyaloperonospora*-infected *Arabidopsis thaliana* Col-0 plants and re-analyze cleavage sites of *AtWNK2* and *AtAED3* by next generation high-throughput sequencing. For both genes, sequencing data revealed only very few 5` ends of mRNAs at the predicted cleavage site, while several orders of magnitude more reads mapped further 3’ downstream of the target sites, indicating rapid degradation of the transcripts. The few remaining 5` ends of mRNAs that mapped at the predicted target sites did not display a prominent peak at the expected position of 10/11. Therefore, we agree with the reviewers that the 5’RACE-PCR does not provide any direct evidence for *Hyaloperonospora* small RNA-directed transcript cleavage. Although being a negative result, we suggest to report these new data in our manuscript in Figure 1—figure supplement 4 for transparency reason, however we have removed all other RACE-PCR data from the Figure 1. Moreover, the RACE-PCR results are now clearly described as inconclusive in the manuscript text as follows:

“We isolated PCR products at the predicted cleavage sizes (Figure 1—figure supplement 4A) for next generation sequencing analysis. […] Therefore, RACE-PCR did not support *Hpa*RNA-guided cleavage of the Arabidopsis target mRNAs.”

We have removed a respective statement from the Abstract section of our manuscript text, accordingly.

2) Supplementary file 2 holds the predicted targets. How conserved are these targeted sequences among Arabidopsis accessions?

We are happy to address this question. We now have analyzed sequence variations at the predicted target sites given in Supplementary file 2 among the sequenced Arabidopsis accessions of the "1001 genome project". Indeed, we detected several SNPs and indels in *Hpa*sRNA target sites in at least one *A. thaliana* accession for 34 out of 49 target genes, and some of the detected mutations most likely impair *Hpa*sRNA-induced silencing. We have now included the information of the target site position within Arabidopsis genes (5`/3` UTRs, CDS), and target site conservation by giving the types and numbers of mutations for each case in the Supplementary file 2. We now further providing a summary of allelic variations comprising mutations that likely disturb target silencing in a new Figure 4—figure supplement 5. We have now described these new findings in the manuscript text as follows: “To gain more information on the conservation of the 34 identified *At*AGO1-associated *Hpa*sRNAs and their 49 predicted plant target sites (Supplementary file 2) we analysed RNA sequence diversity using the *H. arabidopsidis* sequenced genomes of the Noco2, Cala2 and Emoy2 isolates (NCBI BioProject IDs: PRJNA298674; PRJNA297499, PRJNA30969) as well as in 1135 *A. thaliana* accessions published by the 1001 genome project (1001 Genomes Consortium, 2016). Interestingly, all *Hpa*sRNA genes were found by BLASTn search in the three *H. arabidopsidis* isolates with only three allelic variations identified in Emoy2 (Figure 4—figure supplement 5A). On the Arabidopsis target site, we found SNPs and indels in 70 % of all target genes (Supplementary file 2), many of those might impair in the predicted *Hpa*sRNA-induced silencing (Figure 4—figure supplement 5B).” We have joined results of our analysis on sRNA2 conservation among different oomycete species, moving it from the discussion to the result part in the manuscript text.

Other critical points:1) The PR1 and PDF1.2 expression analysis shows no difference. We think the choice of these genes is not optimal as clearly the plant mounts a defense? response in ago1 mutants visible through the trailing necrosis. So, what is activated then?

We understand the reviewers’ concern about expression analysis on *AtPR1* and *AtPDF1.2*. We share the surprise that strong resistance found in *atago1-27* did not result in increased expression of these two defense marker genes. To gain more insights into potential candidate genes associated with the phenotype of trailing necrosis in *atago1-27*, we have now profiled the expression of two reactive burst oxidases, *AtRBOHD* and *AtRBOHF*, in *A. thaliana* Col-0 wild type and *atago1-27* upon infection with *H. arabidopsidis*. We chose these two genes, because enhanced resistance against *H. arabidopsidis* infection accompanied with spread of cell death was previously reported for *atrbohdand atrbohf* knockout mutant plants (Torres et al., 2002). Moreover, a role in limiting salicylic acid-triggered cell death was assigned to *At*RBOHD and *At*RBOHF (Torres et al., 2005). In consistence to published data, *AtRBOHD* and *AtRBOHF* are significantly higher expressed in wild type plants compared to *atago1-27* at 7 days post inoculation. These new data provide a first hint that ROS production pathway might be involved in trailing necrosis in *atago1-27*. We have now included the new data of *AtRBOHD* and *AtRBOHF* gene expression in the new Figure 2—figure supplement 5 and have described these new findings in the manuscript text, as follows:

"To examine plant gene expression related to induced plant cell death, as observed in *ago1* mutants, we measured transcript levels of the two NADPH oxidases *At REACTIVE BURST OXIDASE HOMOLOG* (*AtRBOH*)*B* and *AtRBOHF*. […] These results gave a first hint of a host defence pathway that might be affected due to *At*AGO1-associated *Hpa*sRNAs."

Finding all unknown genes associated to the observed trailing necrosis in *atago1* mutant plants would however require a holistic approach such as an RNA-seq experiment, which we feel goes beyond the scope of this study.

2) We are considering the P. capsici experiments as suboptimal. The images show sporangia and at 48-72hpi most host tissue is likely dead and therefore unable to activate GUS.

We understand the concern of the reviewers that *P. capsici* as a pathogen is rather suboptimal, because it might have killed infected host tissue disabling GUS activation. We would like to explain why we have chosen *P. capsici* in this assay. a) To our understanding, *P. capsici* is – after *H. arabidopsidis* – the second best established oomycete leaf-infecting pathogen of *A. thaliana* with standardized pathogen cultivation and plant inoculation protocols (Wang et al., 2013). b) Although a hemibiotrophic pathogen, *P. capsici* has elongated biotrophic phase of at least two days post inoculation (Wang et al., 2013). We think that images of *P. capsici*-infected *A. thaliana* reporter plants taken at 2-3 days post inoculation (dpi) were at this stage. Supporting this assumption, another Arabidopsis wild type leaf that we infected with *P. capsici* exhibited only few dead plant cells at 2 dpi, while the majority of plant cells remained viable (Author response image 5). The images of *Hpa*sRNA90/*Hpa*sRNA2 ts:Csy4 plants depict not only pathogen sporangia but also hyphae. To make this clearer, we have inserted an arrow that point at hyphae in the improved image in Figure 1—figure supplement 5. We further would like to guide your attention to the Author response image 5, which shows a leaf of the miR164 ts:Csy4 reporter line infected by *P. capsici* at 2 dpi. In this control image, sporangia and hyphae are visible; however, the leaf was stained completely blue due to *At*miR164-triggered GUS activation. Therefore, we are convinced that GUS activity in plant cells that are under infection threat by *P. capsici* at 2 dpi is possible. Based on these observations, we would like to support the results of our Csy4/GUS reporter plants with the inoculation experiments with *P. capsici*.

**Author response image 5. sa2fig5:** Proof of the viability of Arabidopsis cells colonized by *Phytophthora capsici* under inoculation condition used in the Csy4/GUS ck-RNAi reporter-based assays. a) Two representative microscopy images showing Trypan-blue stained Arabidopsis leaves infected with *P. capsici* hyphae (arrows) and oospores (triangle) at 3 dpi. Local plant cell death was started eliciting at infection sites (asterisk); however, most plant cells yet appeared to be still alive. b) A mature leaf of the Csy4/GUS reporter plants including the AtmiR164 target site expressed GUS. The arrow indicates the infection structures of *P. capsici*. GUS activity was stained at 2 dpi. The scale bars represent 50 μm.

3) We are also concerned by the choice of the WNK2 promoter to drive the Csy4 reporter. The authors state that transcript levels of WNK2 are altered in compatible vs incompatible interactions. Is it clear that the transcript levels are altered post-transcriptionally and that the promoter itself shows the same responsiveness in different infection scenarios?

We thank the reviewers for pointing out the remaining ambiguity of the intentional choice of the *AtWNK2* promoter in the reporter construct. To our opinion, *Hyaloperonospora* infection-specific GUS activation must be attributed to posttranscriptional regulation in the Arabidopsis reporter line, as the same *AtWNK2* promoter was used for all reporter constructs including the negative controls. Important to note, all transgenic plant reporter lines displayed full compatibility with *H. arabidopsidis*, we therefore exclude the possibility that differential promoter activation due to incompatible interaction could result in GUS activation only in the native target site version, but not in versions carrying the *At*miR164 target site or scrambled sequences. We have now included an explanation in the manuscript text that all reporter constructs were under the control of the same *proAtWNK2* promoter and that all transgenic reporter plant lines were fully compatible with *H. arabidopsidis*. Therefore, we concluded that the observed GUS activity induced by *H. arabidopsidis* in plants expressing Csy4 fused to *Hpa*sRNA2/*Hpa*sRNA90 target sites was neither due to target sequence-unspecific regulation of Csy4 or GUS nor pathogen-triggered regulation of the *AtWNK2* promoter.

4) Have the authors addressed whether the Hpa sRNAs could also target Hpa transcripts or are they exclusive to plant transcripts? I could not find such data.

*Hpa*sRNAs listed in Supplementary file 2 are derived from non-coding regions of the *Hyaloperonospora* genome. Indeed, it is possible that they could also be capable of regulating endogenous *Hyaloperonospora* mRNAs in trans. However, sRNA-induced gene silencing in oomycetes has, to our knowledge, only been suggested *in cis* for instance in co-expressional silencing of transposons and effector genes in *Phytophthora* (Jia et al., 2017; Qutob et al., 2013). We would here be only able to provide *in silico* prediction of *Hyaloperonospora* mRNA targets based on the assumption that oomycete small RNAs function like plant small RNAs. We decided to not include this analysis in our study, because we feel that such analysis would be premature and would not provide any significantly novel insights into *Hpa*sRNA-induced endogenous gene regulation to the readership. However, if reviewers and the editor feel that such data could be useful to report, we are happy to provide an analysis on this subject.

5) Point 4 in the previous points from reviewers is still valid and, in our view, using bacteria as an additional system is debatable.

We understand that using a bacterial pathogen in this assay is debatable. We have shifted these data into the figure supplement. To strengthen the result of this assay, we now provide additional data that based on two new transgenic Arabidopsis plant lines expressing either a STTM complementary to a random, scrambled sRNA sequence or complementary to a *Hyaloperonospora* rRNA-derived sRNA sequence. For both constructs, at least 5 independent T1 lines were challenged with *H. arabidopsidis*, and we did not detect any case of trailing necrosis. These results further support that expression of STTMs did not stimulate plant immunity *per se*, but blocked *Hpa*sRNA2, *Hpa*sRNA30 and *Hpa*sRNA90 activity, which resulted in reduced virulence of *H. arabidopsidis*. The new data are now included in Figure 3D and are described in the manuscript text as follows:

"We also cloned STTMs against an rRNA-derived *Hpa*sRNA as well as against a random scrambled sequence for expression in Arabidopsis. These two types of control STTMs did not exhibit trailing necrosis in at least 5 independent T1 transgenic lines upon *H. arabidopsidis* inoculation (Figure 3D)."

**References:**

Cai Q, He B, Weiberg A, Buck AH, Jin H. 2019. Small RNAs and extracellular vesicles: New mechanisms of cross-species communication and innovative tools for disease control. *PLoS Pathog* 15:e1008090. doi:10.1371/journal.ppat.1008090

Chandran D, Inada N, Hather G, Kleindt CK, Wildermuth MC. 2010. Laser microdissection of Arabidopsis cells at the powdery mildew infection site reveals site-specific processes and regulators. *Proc Natl Acad Sci* 107:460–465. doi:10.1073/pnas.0912492107

Hacquard S, Delaruelle C, Legué V, Tisserant E, Kohler A, Frey P, Martin F, Duplessis S. 2010. Laser capture microdissection of uredinia formed by *Melampsora laricipopulina* revealed a transcriptional switch between biotrophy and sporulation. *Mol Plant Microbe Interact* 23:1275–1286. doi:10.1094/MPMI-05-10-0111

Knoth C, Eulgem T. 2008. The oomycete response gene *LURP1* is required for defense against *Hyaloperonospora parasitica* in *Arabidopsis thaliana*. *Plant J* 55:53–64. doi:10.1111/j.1365-313X.2008.03486.x

van Wees S. 2008. Phenotypic analysis of *Arabidopsis* mutants: Trypan blue stain for fungi, oomycetes, and dead plant cells. *Cold Spring Harb Protoc* 2008:pdb.prot4982pdb.prot4982. doi:10.1101/pdb.prot4982

Wang Y, Bouwmeester K, van de Mortel JE, Shan W, Govers F. 2013. A novel Arabidopsis-oomycete pathosystem: differential interactions with Phytophthora capsici reveal a role for camalexin, indole glucosinolates and salicylic acid in defence. Plant Cell Environ 36:1192–1203. doi:10.1111/pce.12052